Letter

# Somatic mutations in facial skin from countries of contrasting skin cancer risk

Charlotte King [1], Joanna C. Fowler[1], Irina Abnizova[1], Roshan K. Sood [1], Michael W. J. Hall [1,2], Ildikó Szeverényi[3,6], Muly Tham[3], Jingxiang Huang[3], Stephanie Ming Young[4], Benjamin A. Hall [5], E. Birgitte Lane [3] & Philip H. Jones [1,2] ✉

The incidence of keratinocyte cancer (basal cell and squamous cell carcinomas of the skin) is 17-fold lower in Singapore than the UK[1–3], despite Singapore receiving 2–3 times more ultraviolet (UV) radiation[4,5]. Aging skin contains somatic mutant clones from which such cancers develop[6,7]. We hypothesized that differences in keratinocyte cancer incidence may be reflected in the normal skin mutational landscape. Here we show that, compared to Singapore, aging facial skin from populations in the UK has a fourfold greater mutational burden, a predominant UV mutational signature, increased copy number aberrations and increased mutant TP53 selection. These features are shared by keratinocyte cancers from high-incidence and low-incidence populations[8–13]. In Singaporean skin, most mutations result from cell-intrinsic processes; mutant NOTCH1 and NOTCH2 are more strongly selected than in the UK. Aging skin in a high-incidence country has multiple features convergent with cancer that are not found in a low-risk country. These differences may reflect germline variation in UV-protective genes.

The incidence of many cancers varies substantially worldwide, reflecting genetic differences between populations and their environmental exposures. This is well illustrated by keratinocyte cancers, where incidence varies 140-fold globally[14]. Keratinocyte cancer risk increases with an individual's cumulative ultraviolet (UV) exposure that depends on age, outdoor work, sunbathing, use of tanning beds[15–20] and phenotypes such as freckles, low levels of skin pigmentation and poor tan response[21]. The most strongly associated keratinocyte cancer risk loci are found near to pigmentation genes such as HERC2, OCA2, MC1R and IRF4, which have much greater allele frequencies in high-risk populations such as in the UK than in low-risk populations in East Asia[22].

The intensity of UV reaching the Earth's surface is quantified using the linearly scaled UV index[23]. The average daily maximum UV index in Singapore is 8 (https://www.nea.gov.sg/) compared with 3 in the UK (https://uk-air.defra.gov.uk/data/uv-data). Despite this,

the age-adjusted incidence of keratinocyte cancers is 17-fold lower in Singapore than in the UK (Fig. 1a)[24]. Furthermore, keratinocyte cancer incidence has risen rapidly in the UK but not in Singapore.

The aging epidermis of skin from donors from the UK consists of a dense patchwork of somatic clones, with mutant NOTCH1, NOTCH2, NOTCH3, TP53 and FAT1 under positive selection[6,7]. These genes are commonly mutated in keratinocyte cancers, particularly cutaneous squamous cell carcinoma (cSCC)[8,11], suggesting that these tumors develop from such mutant clones. To date, little is known of how this somatic landscape varies across human populations.

In this study, we characterized the somatic clones in histologically normal aging eyelid epidermis of five donors from Singapore compared with published sequencing data[7] from six similar donors from the UK (Fig. 1b). Donor sex and age were similar in both countries (mean donor age Singapore = 62 years, UK = 68 years; Table 1).

[1]Wellcome Sanger Institute, Hinxton, UK. [2]Department of Oncology, University of Cambridge, Hutchinson Research Centre, Cambridge Biomedical Campus, Cambridge, UK. [3]Present address: Skin Research Institute of Singapore and Institute of Medical Biology, Agency for Science, Technology and Research (A*STAR), Singapore, Singapore. [4]Department of Ophthalmology, National University Hospital, Singapore, Singapore. [5]Department of Medical Physics and Biomedical Engineering, University College London, London, UK. [6]Institute of Aquaculture and Environmental Safety, Georgikon Campus, Hungarian University of Agricultural and Life Sciences, Keszthely, Hungary. ✉e-mail: pj3@sanger.ac.uk

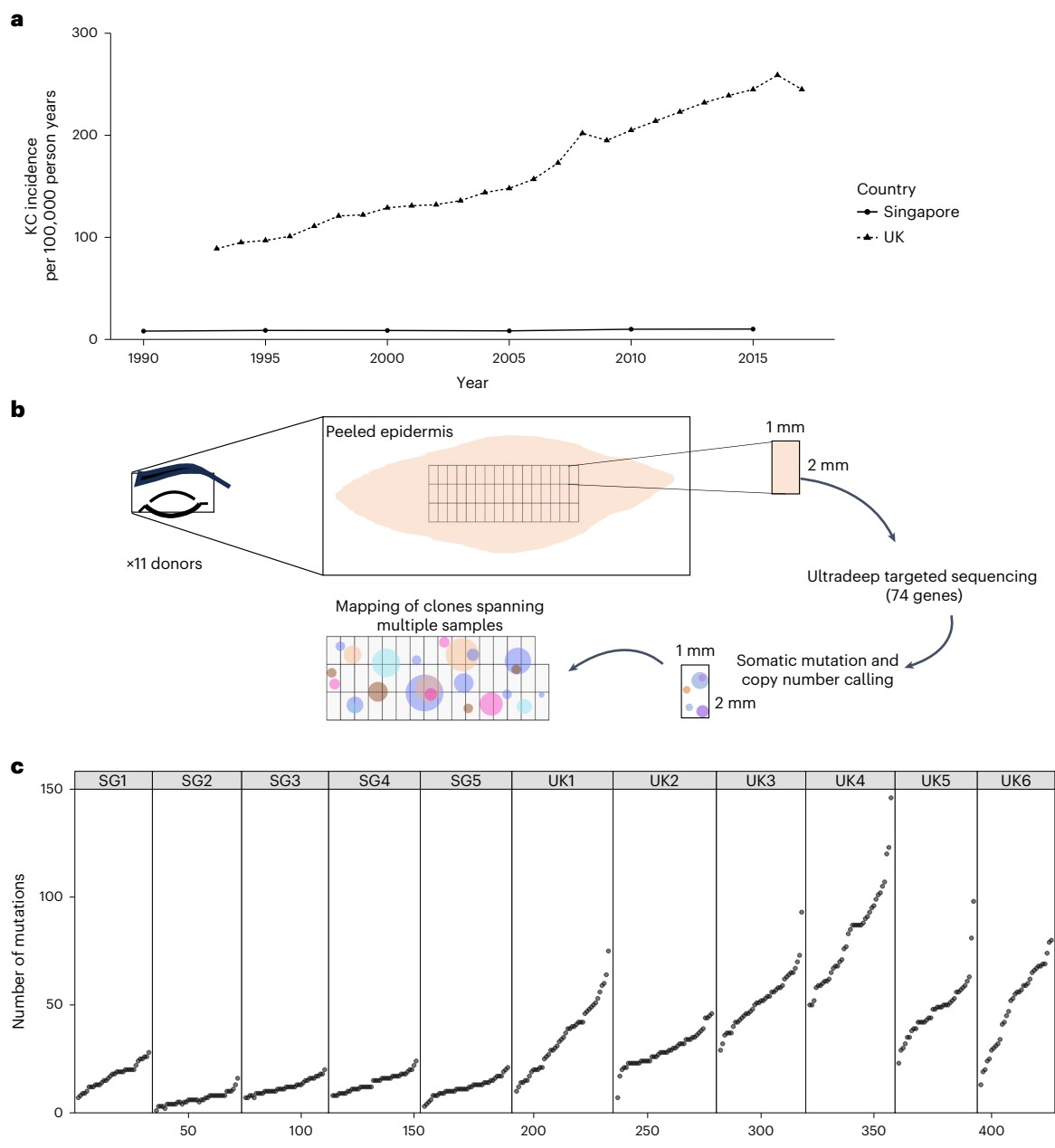

**Fig. 1 | Sampling of facial skin across two countries with contrasting skin cancer risk. a**, Age-standardized incidence rates per 100,000 person years for keratinocyte cancers (KC) in the UK and Singapore. Data were collated from Cancer Research UK and the Singapore Cancer Registry[1,2]. **b**, Sampling method to allow mapping of clones spanning multiple samples of epidermis. **c**, Number of mutations detected per 2-mm² sample of epidermis (n = 428) across sequencing of 74 genes per donor.

A total of 13,850 mutations was detected across all samples (Fig. 1c). The number of mutations per sample varied from 0.5 to 73 clones per mm² (mean = 16.2 clones per mm²; Supplementary Table 1). We estimated the mean genome-wide burden per donor to be four-fold higher in the UK (6.3 mutations per Mb) compared to Singapore (1.6 mutations per Mb; Fig. 2a, Extended Data Fig. 1a and Supplementary Table 2).

Across all donors, 10,311 single-base substitutions (SBS) were detected, of which 66% were C>T changes (Fig. 2b). Mutational signature analysis identified six reference signatures[25] SBS1, SBS5 and SBS7a–d (Supplementary Table 3). SBS1 and SBS5 are associated with tissue aging while SBS7a–d are due to UV lesions and their repair[25]. Mutational signatures differed between countries (Fig. 2c). Most (64%) SBS in Singaporean donors were attributed to SBS1 and SBS5, but in the UK, most (66%) were attributed to SBS7a–d. The proportion of substitutions attributed to UV were positively correlated with genome-wide burden estimates per donor (Fig. 2d).

The number of both SBS1 and SBS5 mutations per mm² was increased in UK skin compared with Singaporean skin (Extended Data Fig. 1b,c). In epithelial cells, the proportion of the SBS1 mutation was previously correlated with the cumulative number of cell divisions in a tissue. In this study, in both countries, we found increased SBS1 in larger clones (Extended Data Fig. 1d). These differences may reflect the effects of UV, which increases the rate of proliferation and proportion of dividing cells in the epidermis and can drive mutant clone expansion[26].

## Table 1 | Donor demographics

| Donor | Site | Age | Sex | Occupation | Fitzpatrick score | Smoker | Total area sampled (mm²) | Genome-wide burden estimate (mutations per Mb) |
|---|---|---|---|---|---|---|---|---|
| SG1 | Eyelid | 68–71 | F | Indoor | 3 | No | 68 | 3.75 |
| SG2 | Eyelid | 28–31 | F | Indoor | 4 | No | 78 | 0.62 |
| SG3 | Eyelid | 72–75 | M | Outdoor | 4 | No | 76 | 0.74 |
| SG4 | Eyelid | 76–79 | M | – | 3 | No | 80 | 1.31 |
| SG5 | Eyelid | 56–59 | M | Mixed | 4 | Ex-smoker | 80 | 1.34 |
| UK1 | Eyelid | 60–63 | M | Indoor | 2 | Yes | 88 | 4.24 |
| UK2 | Eyelid | 76–79 | F | Indoor | 2 | – | 90 | 4.13 |
| UK3 | Eyebrow | 64–67 | F | Indoor | 2 | Ex-smoker | 78 | 11.9 |
| UK4 | Eyebrow | 76–79 | M | Mixed | – | – | 78 | 4.87 |
| UK5 | Eyebrow | 72–75 | M | Outdoor | 2 | No | 72 | 5.50 |
| UK6 | Eyebrow | 48–51 | F | Indoor | – | Yes | 68 | 7.35 |

UK1–6=PD37619, PD43995, PD43991, PD43992, PD43994 and PD43996, respectively. SG, Singapore.

A total of 561 double-base substitutions (DBS), 388 deletions and 94 insertion events were detected. Over eight times more DBS were called in UK skin compared to Singapore (Extended Data Fig. 1e; UK mean = 1.08 DBS per mm², Singapore mean = 0.126 DBS per mm²). Most DBS (85%) were CC>TT substitutions and showed transcriptional strand bias consistent with UV damage (transcribed/untranscribed = 52.7). There was no significant difference in the number of insertions and deletions (indels) detected according to country (Extended Data Fig. 1f; UK mean = 0.72 indels per mm², Singapore mean = 0.37 indels per mm²).

We detected copy number aberrations (CNAs) in 32 samples, of which 27 were independent events (Supplementary Table 4). Seventy-eight percent of CNAs detected were loss of heterozygosity (LOH) at the *NOTCH1* locus on 9q, which is consistent with mutant *NOTCH1* being the most common driver of clonal expansion in normal skin[6,7]. Other CNAs included two *TP53* LOH events, one *NOTCH4* duplication and an LOH event at each of *NOTCH4*, *FGFR2* and *RB1*. CNAs were detected in 13% of UK and just 1.0% of Singaporean samples.

We found that *NOTCH1*, *FAT1*, *TP53*, *NOTCH2*, *NOTCH3*, *ARID2* and *AJUBA* harbored a disproportionately high number of non-synonymous mutations relative to synonymous mutations (d*N*/d*S* ratio, *q* < 0.01), suggesting that protein-altering mutations in these genes drive clonal expansions (Extended Data Fig. 2a and Supplementary Table 5), which is consistent with previous studies[6,7]. A comparison of the d*N*/d*S* ratio per gene by country found *TP53* non-synonymous mutations overrepresented and *NOTCH1* and *NOTCH2* non-synonymous mutations underrepresented in UK skin compared to Singaporean skin (Fig. 3a–c). We estimated the percentage of tissue mutant for *NOTCH1*, *NOTCH2*, *FAT1* and *TP53* (Fig. 3d and Supplementary Table 6). We found a fourfold increase in tissue mutant for *FAT1* (UK mean = 11%, Singapore mean = 2.6%), which is consistent with a fourfold higher mutation burden in UK skin compared to Singapore. However, we only observed a twofold increase in tissue mutant for *NOTCH1* (UK mean = 24%, Singapore mean = 12%) and no significant difference in the percentage of tissue mutant for *NOTCH2* (UK mean = 6.4%, Singapore mean = 7.7%). Thus, *NOTCH1* and *NOTCH2* mutants are less able to colonize UK skin than expected.

For *TP53*, we observed a threefold increase in the percentage of mutant tissue in the UK (UK mean = 14.5%, Singapore mean = 4.9%; Fig. 3d). However, this is only of borderline significance due to a large *TP53* mutant clone that spans 16 samples of donor SG1. When we removed this single clone, we found a ninefold difference (UK mean = 14.5%, Singapore mean = 1.7%). We concluded that mutant *TP53* is probably a better competitor than mutant *NOTCH1* and *NOTCH2* in UK skin.

One explanation for the differences in selection we observed in Singaporean skin is that mutant *NOTCH1* and *NOTCH2* clones are relatively strong competitors because, in a less mutated tissue, they are more likely to be competing against nonmutant cells. We identified further lines of evidence in support of this hypothesis. First, UK skin is saturated with mutant cells. We estimated that an average of 94% of cells in UK donors have a protein-altering mutation in at least one of the sequenced genes, compared to 50% of cells in Singaporean donors (Fig. 3e and Supplementary Table 7). Second, mean clone size was larger in Singaporean skin (0.037) than in UK skin (0.029; Fig. 3f and Supplementary Table 1). Third, *NOTCH1* and *NOTCH2* mutant clones were larger in Singapore than in the UK (Extended Data Fig. 2b). This contrasts with *TP53* mutant clones, which did not significantly differ in size according to country, suggesting that they have a strong relative fitness in both environments. These observations may be reconciled with a simple model in which a high burden of mutant clones restricts clone size through competition for the space available within the tissue (Supplemetary Videos 1 and 2)[27].

Protein-altering *TP53* mutations differed according to country. The two most frequent codon changes in UK skin were *TP53*^R248W and *TP53*^R282W (3.2 and 2.5 mutations per cm², respectively). Both have gain-of-function (GOF) properties that can lead to chromosomal instability and R248W is the most frequent codon change in keratinocyte cancers[26,28–30]. We did not observe any mutations at either of these codons in the Singaporean samples (Extended Data Fig. 2c). Even after adjusting for C>T/CC>TT burden across *TP53* by country, we would still expect to observe approximately three mutations at codons R248 and R282 across all Singaporean samples (Methods; bootstrapping *P* < 0.001). This suggests that the underrepresentation of mutations at these codons in Singaporean skin may not be due to differences in mutational signature alone.

Other activating mutations called in the UK samples included multiple *FGFR3* mutants: K652M (*n* = 2), G382R (*n* = 2), R248C (*n* = 1), S249C (*n* = 1), G372C (*n* = 1) and Y375C (*n* = 1). *FGFR3* GOF mutations drive the formation of the benign keratinocyte tumor seborrheic keratosis[31]. Oncogenic activating mutations were also found: *KRAS*^G12D (*n* = 1); *NRAS*^Q61L (*n* = 1); and *HRAS*^E143K (*n* = 2) and *HRAS*^G12D (*n* = 1). In Singaporean samples, we observed a single occurrence of an oncogenic activating mutation: *KRAS*^G12V.

To gain an insight into differences in genetic background that may exist between donors, we genotyped individuals for single-nucleotide polymorphisms (SNPs) associated with altered risk for keratinocyte cancer (Methods, Fig. 4 and Supplementary Table 8). At 36 risk loci, we found differences in donor genotype according to country. Many were associated with genes linked to pigmentation, such as *SLC45A2*, *IRF4*,

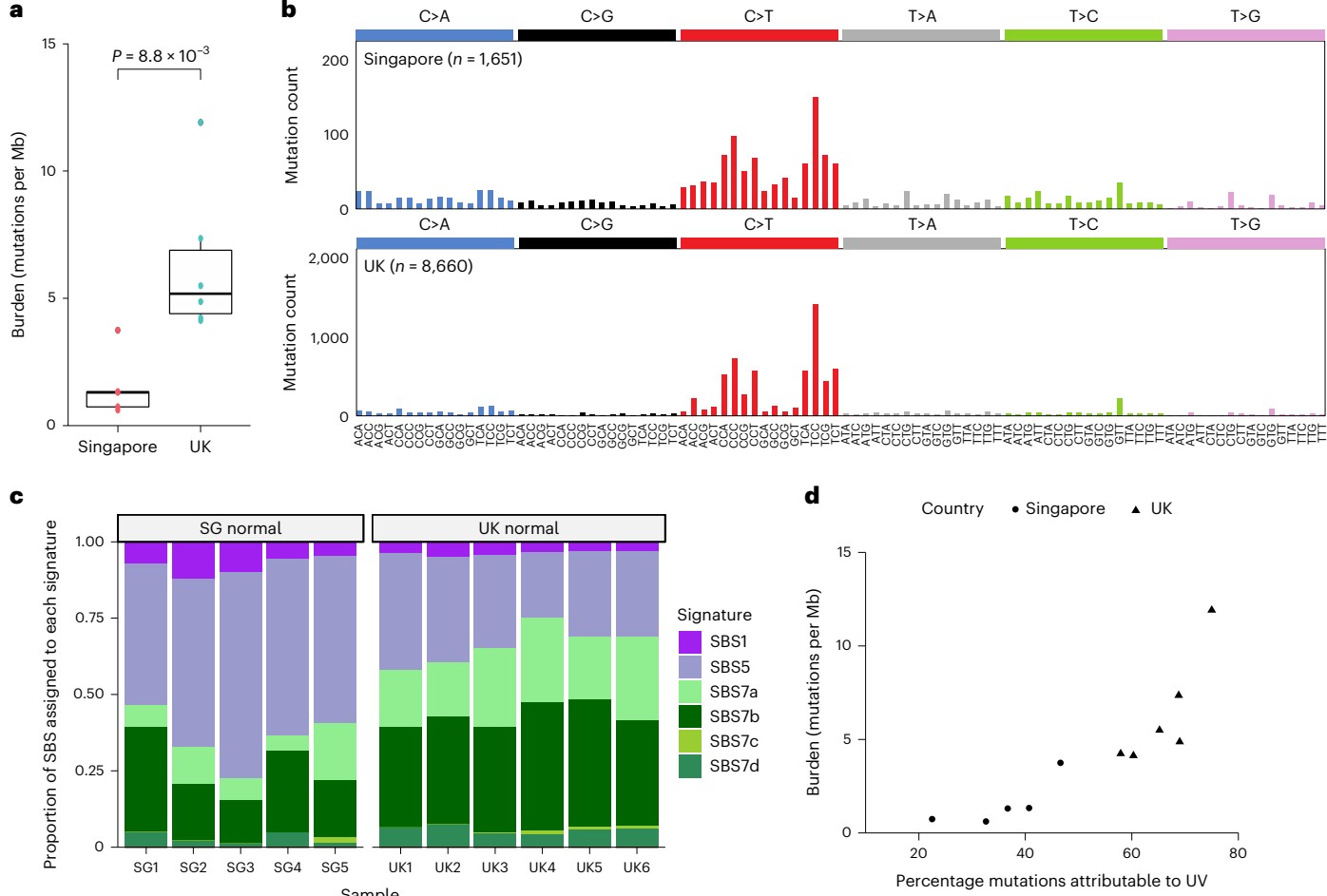

**Fig. 2 | Mechanisms of mutagenesis differ by country. a**, Estimates of genome-wide mutation burden per donor according to country (*n* = 11 donors, two-sided *t*-test: *P* = 8.8 × 10⁻³). Tukey box plot where the lower and upper hinges represent the first and third quartile, the center is the median and the outliers are more than 1.5× the interquartile range (IQR). **b**, Trinucleotide context for SBS in skin from UK and Singaporean donors. **c**, Proportion of SBS assigned to each signature per donor (purple, aging; green, UV). Signature contributions are clustered significantly according to country (Methods; unsupervised hierarchical clustering with accuracy obtained through bootstrapping: a.u. = 0.99999, *P* < 1 × 10⁻⁵). **d**, Proportion of SBS attributed to UV damage was positively correlated with burden estimates per donor (Pearson's *r* = 0.87, Spearman rank = 0.95).

*BNC2*, *OCA2* and *HERC2* (https://genetics.opentargets.org). For example, rs12203592 alters *IRF4* levels and expression of the pigmentation enzyme encoded by *TYR*[32]. rs4778210, positively selected in Singaporeans[33], is adjacent to *OCA2*, encoding a melanosomal anion channel that is essential for melanin synthesis[34]. These findings are consistent with the marked difference in UV mutational burden between countries. However, we also observed differences in non-pigmentation-related SNPs. Keratinocyte cancer risk is strongly linked to immunosuppression and is also associated with inflammatory diseases[21]. rs2111485 is associated with inflammatory and autoimmune diseases, such as psoriasis and inflammatory bowel disease, while rs12129500 and rs2243289 affect the expression of *IL6R* and *IL4*, respectively (https://genetics.opentargets.org). We found that four of the Singaporean donors have a large introgression at chromosome 3p21.31 (ref. 33). This region is under positive natural selection in East Asia and contains the skin tumor suppressor gene *RASSF1* (ref. 35) and the UV-induced genes *HYAL1* and *HYAL2* (ref. 36). Other keratinocyte cancer risk SNPs were linked to genes with diverse functions.

Most keratinocyte cancer sequencing studies have been performed in high-incidence countries and may not reflect tumors from low-incidence populations. We therefore analyzed the whole-exome sequencing data of cheek cSCCs taken from 19 South Korean patients[12].

The age-standardized incidence of cSCC in South Korea (2.38 per 100,000)[37] is comparable to that of Singapore (2.2 per 100,000)[24]. After genotyping the 19 patients for differential risk loci (Fig. 4), we found the typical Korean genotype most similar to that of the Singaporean donors (Supplementary Table 9).

Analysis of South Korean cSCC for mutational signatures identified nine reference signatures: SBS1, SBS5, SBS7a–d, APOBEC-associated signatures SBS2 and 13, and, in one sample (MP7), the defective DNA repair signature SBS15 (Fig. 5a and Supplementary Table 3)[25]. In all but two tumors (MP7 and W2), most mutations were caused by UV (Fig. 5b). Indel burden was several fold higher in samples MP7 and W2 (Fig. 5c), which is consistent with an alternative mechanism of mutagenesis in these cases. Analysis of mutant gene selection found *TP53*, *NOTCH1* and the cell cycle regulating tumor suppressor *CDKN2A.p16INK4a* as positively selected (Fig. 5d and Supplementary Table 5), which is consistent with previous studies[8–10]. In conclusion, cSCCs show convergent genomic features, whether from high-risk or low-risk populations.

Overall, we conclude that differences in keratinocyte cancer incidence between the UK and Singapore are reflected in the somatic mutational landscape of aging facial skin. In comparison to Singapore, UK skin shares multiple features with keratinocyte cancer, including a high mutational burden, a predominant UV mutational signature, increased

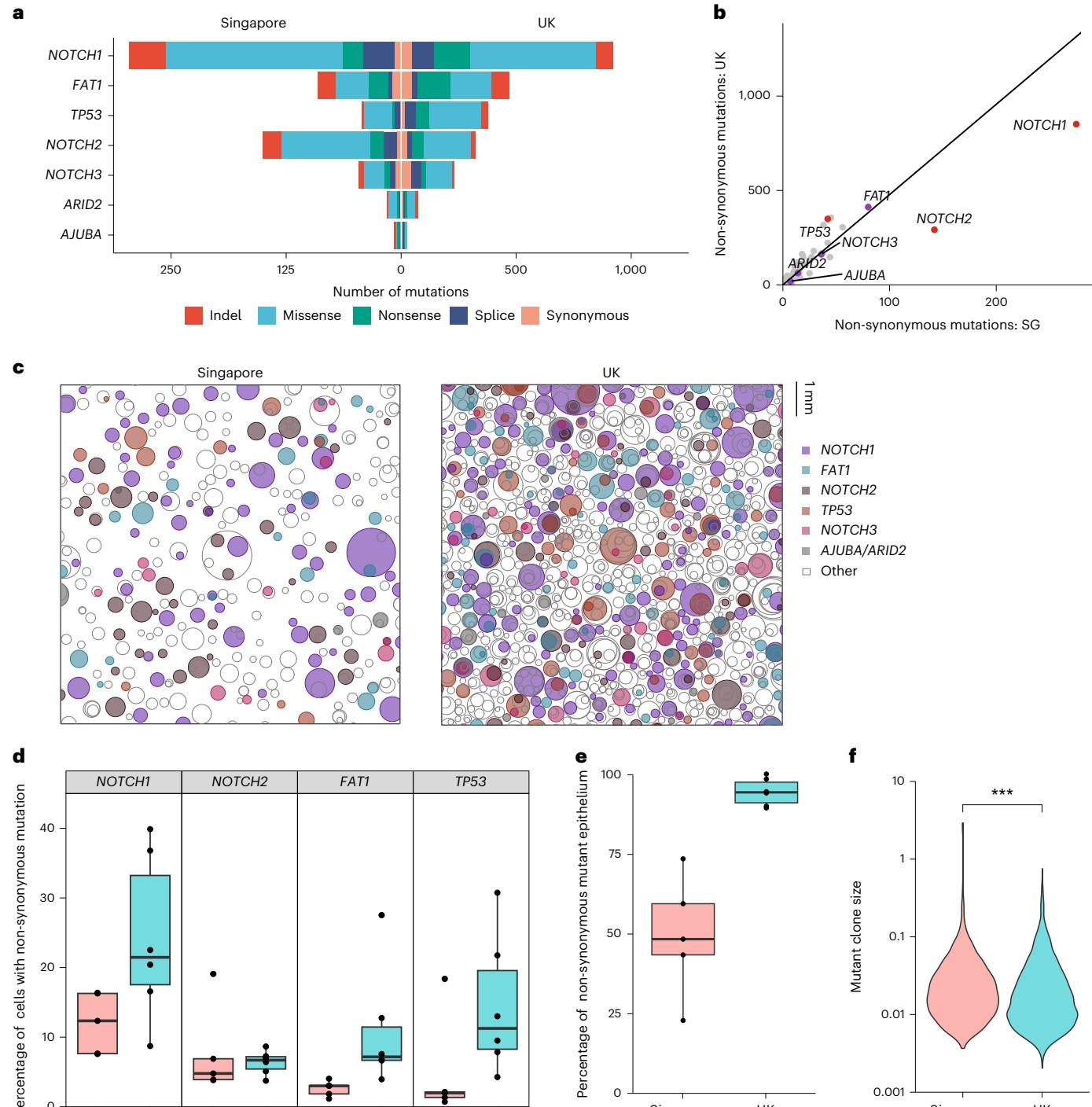

**Fig. 3 | Clonal selection and competition differ according to country.**
**a**, Number of mutations of each consequence for positively selected genes (dNdScv: $q < 0.01$) according to country. *ARID2* is not significantly positively selected in the Singapore samples. **b**, Plot of non-synonymous mutations per gene in Singapore versus UK samples. Gradient of line: total number of non-synonymous mutations in the UK/Singapore = 6,846/1,432. Positively selected genes (purple) are labeled. Red indicates positively selected genes with a significant (one-sided likelihood ratio test: $P_{adj} < 0.001$) difference in d$N$/d$S$ ratio according to country, after accounting for global differences. **c**, A representation of protein-altering mutations in 1 cm² of skin from donors from Singapore and the UK. Samples were randomly selected and mutations are displayed as circles, randomly distributed in the space. Sequencing data, including copy number, were used to infer the size and number of clones and, where possible, the nesting of subclones. Otherwise, subclones are nested randomly. **d**, Estimated

percentage of cells with at least one non-synonymous mutation per positively selected gene, according to country ($n = 11$ donors, samples with known CNA removed). Tukey box plot where the lower and upper hinges represent the first and third quartile, the center represents the median and the outliers are more than 1.5× the IQR. Two-sided Wilcoxon signed-rank test: $P_{adj} = 0.03$ (*NOTCH1*), 0.66 (*NOTCH2*), $8.7 \times 10^{-3}$ (*FAT1*) and 0.05 (*TP53*) with Holm multiple testing correction. **e**, Estimated percentage of cells with at least one non-synonymous mutation across 74 genes according to country (Methods; $n = 11$ donors, two-sided Wilcoxon signed-rank test: $P = 4.3 \times 10^{-3}$). Tukey box plot where the lower and upper hinges represent the first and third quartile, the center represents the median and the outliers more than 1.5× the IQR. **f**, Violin plot comparing clone size distributions (summed variant allele fraction) per mutation according to country. UK donor mutations had a lower mean clone size (two-sided Welch's $t$-test: $P = 4.27 \times 10^{-14}$).

| CHR | POS | REF | ALT | SNP ID | Gene | EA freq | GBR freq | Panel | Suggested mechanism | SG1 | SG2 | SG3 | SG4 | SG5 | UK1 | UK2 | UK3 | UK4 | UK5 | UK6 |
|---|---|---|---|---|---|---|---|---|---|---|---|---|---|---|---|---|---|---|---|---|
| 1 | 17749504 | G | A | rs730153 | RCC2 | 0.17 | 0.32 | KC | Carcinogenesis | 0/0 | 0/0 | 0/0 | 0/0 | 0/0 | 0/0 | 0/1 | 0/0 | 0/1 | 0/0 | 0/1 |
| 1 | 154423764 | T | C | rs12129500 | IL6R | 0.43 | 0.59 | KC | Inflammation/immune response | 0/0 | 0/1 | 0/0 | 0/1 | 0/0 | 1/1 | 1/1 | 0/0 | 1/1 | 1/1 | 0/1 |
| 2 | 163110536 | A | G | rs2111485 | FAP | 0.17 | 0.62 | KC | Carcinogenesis | 0/0 | – | 0/0 | 0/0 | 0/0 | 1/1 | 0/1 | 0/1 | 1/1 | 1/1 | 0/1 |
| 3 | 71483084 | G | A | rs62246017 | FOXP1 | 0.00 | 0.29 | KC | Carcinogenesis | – | – | 0/0 | 0/0 | 0/0 | 0/0 | – | 1/1 | 0/1 | 0/0 | 1/1 |
| 5 | 33951693 | C | G | rs16891982 | SLC45A2 | 0.01 | 0.98 | KC | Pigmentation | 0/0 | 0/0 | 0/0 | 0/0 | 0/0 | 1/1 | – | – | 1/1 | 1/1 | 1/1 |
| 6 | 396321 | C | T | rs12203592 | IRF4 | 0.00 | 0.18 | KC, Tan | Inflammation/immune response | – | 0/0 | 0/0 | 0/0 | 0/0 | – | – | – | 1/1 | 0/0 | 0/1 |
| 6 | 427972 | C | T | rs9405665 | IRF4 | 0.25 | 0.41 | Tan | Pigmentation | 0/0 | 0/0 | 0/0 | 0/1 | 0/0 | 0/1 | 0/1 | 0/1 | 0/1 | 0/1 | 0/1 |
| 9 | 16913836 | T | A | rs11445081 | BNC2 | 0.00 | 0.56 | KC, Tan | Pigmentation | 0/0 | 0/0 | 0/0 | 0/0 | 0/0 | 1/1 | 0/0 | 0/1 | 1/1 | 1/1 | 0/1 |
| 9 | 139356987 | T | C | rs57994353 | SEC16A | 0.06 | 0.29 | KC | Inflammation/immune response | 0/0 | 0/0 | 0/0 | 0/0 | – | 0/1 | 1/1 | 0/1 | 0/0 | 0/0 | 0/1 |
| 11 | 47430192 | T | TCGC | rs551004316 | SLC39A13 | 0.26 | 0.34 | Tan | ECM | 0/0 | 0/0 | 0/0 | 0/0 | 0/0 | 0/1 | 0/0 | 0/0 | 0/0 | 0/0 | 0/0 |
| 11 | 64107735 | G | A | rs663743 | PPP1R14B | 0.16 | 0.34 | KC | Carcinogenesis | 0/0 | 0/0 | 0/0 | 0/0 | 0/0 | 0/0 | 0/0 | 0/1 | 0/0 | 0/1 | 0/0 |
| 11 | 95311422 | T | C | rs4409785 | SESN3 | 0.10 | 0.21 | KC | Oxidative stress, autophagy | – | – | 0/0 | 0/0 | – | – | 0/1 | – | – | 0/1 | – |
| 13 | 100041738 | G | C | rs7335046 | UBAC2 | 0.67 | 0.84 | KC | Cell proliferation | 0/0 | 0/0 | 0/0 | 0/0 | 0/0 | 0/0 | 0/1 | 0/0 | 0/1 | 0/1 | 0/0 |
| 13 | 113537448 | G | A | rs3742233 | ATP11A | 0.24 | 0.49 | Tan | Pigmentation | 0/0 | 0/0 | 0/1 | 0/0 | 0/0 | 0/0 | 0/1 | 0/1 | 0/1 | 0/1 | 0/0 |
| 15 | 28,410,491 | C | T | rs12916300 | HERC2 | 0.00 | 0.81 | KC | Pigmentation | 0/0 | 0/0 | 0/0 | 0/0 | 0/0 | 0/0 | – | 1/1 | 1/1 | 1/1 | 1/1 |
| 15 | 28530182 | C | T | rs1667394 | HERC2 | 0.27 | 0.88 | Tan | Pigmentation | 0/0 | – | 0/0 | 0/1 | 0/0 | 0/0 | 1/1 | – | 1/1 | – |  |
| 20 | 36017340 | T | G | rs754626 | SRC | 0.02 | 0.32 | KC | Carcinogenesis | 0/0 | 0/0 | 0/0 | 0/0 | 0/0 | 0/1 | 0/1 | 1/1 | 0/0 | 0/1 | 0/0 |
| 21 | 43429646 | G | A | rs399907 | C2CD2 | 0.15 | 0.76 | Tan | Pigmentation | 0/0 | 0/0 | 0/0 | 0/0 | 0/0 | 0/1 | 0/0 | 0/1 | 0/1 | 0/1 | 1/1 |
| 2 | 216318927 | G | C | rs1250219 | FN1 | 0.94 | 0.16 | SG | ECM | – | – | 1/1 | 1/1 | – | 0/0 | 0/0 | – | 0/0 | – | 0/0 |
| 3 | 50311900 | C | T | rs2071203 | HYAL region | 0.62 | 0.01 | SG | tumor suppression, ECM | 1/1 | 1/1 | 1/1 | 1/1 | – | 0/0 | 0/0 | 0/0 | 0/0 | 0/0 | 0/0 |
| 3 | 50323172 | C | T | rs12494414 | HYAL region | 0.63 | 0.01 | SG | tumor suppression, ECM | 1/1 | 0/0 | 1/1 | 1/1 | 0/0 | 0/0 | 0/0 | 0/0 | 0/0 | 0/0 | 0/0 |
| 3 | 50334231 | T | A | rs2269432 | HYAL region | 0.62 | 0.01 | SG | tumor suppression, ECM | 1/1 | 1/1 | 1/1 | 1/1 | 0/0 | 0/0 | 0/0 | 0/0 | 0/0 | 0/0 | 0/0 |
| 3 | 50352200 | T | G | rs11130248 | HYAL region | 0.62 | 0.01 | SG | tumor suppression, ECM | 1/1 | 1/1 | 1/1 | 1/1 | 0/0 | 0/0 | 0/0 | 0/0 | 0/0 | 0/0 | 0/0 |
| 3 | 50355730 | T | G | rs35455589 | HYAL region | 0.62 | 0.01 | SG | tumor suppression, ECM | 1/1 | 1/1 | 1/1 | 0/0 | 0/0 | 0/0 | 0/0 | 0/0 | – | 0/0 | 0/0 |
| 3 | 50363771 | G | A | rs2236946 | HYAL region | 0.62 | 0.01 | SG | tumor suppression, ECM | 1/1 | 1/1 | 1/1 | 1/1 | 0/0 | 0/0 | 0/0 | 0/0 | 0/0 | 0/0 | 0/0 |
| 3 | 50378176 | T | G | rs4688725 | HYAL region | 0.62 | 0.01 | SG | tumor suppression, ECM | 1/1 | 1/1 | 1/1 | 1/1 | 0/0 | 0/0 | 0/0 | 0/0 | 0/0 | 0/0 | 0/0 |
| 4 | 99983312 | T | C | rs1238741 | METAP1 | 0.81 | 0.10 | SG | – | 1/1 | 1/1 | 1/1 | 1/1 | 0/1 | 0/0 | 0/0 | – | 0/0 | – | 0/0 |
| 5 | 132018132 | A | G | rs2243289 | IL4 | 0.78 | 0.12 | SG | Inflammation/immune response | 1/1 | 1/1 | 1/1 | 0/0 | 1/1 | 0/0 | – | 0/1 | 0/0 | 0/1 | 0/0 |
| 5 | 149196682 | T | G | rs109077 | – | 0.38 | 0.29 | KC | Pigmentation | 0/0 | 0/0 | 0/1 | 0/1 | 0/0 | 0/0 | 0/0 | 0/1 | 0/0 | 0/0 | 0/0 |
| 6 | 392690 | CG | C | rs11308001 | IRF4 | 0.78 | 0.51 | Tan | Pigmentation | 0/0 | 0/1 | 0/1 | 0/1 | 0/1 | 0/0 | 0/0 | 0/0 | 0/0 | 0/1 | 0/0 |
| 6 | 21494984 | G | A | rs6935474 | SOX4 | 0.94 | 0.31 | SG | Carcinogenesis | 1/1 | 1/1 | 1/1 | 1/1 | 1/1 | – | 0/0 | 0/1 | 0/0 | 1/1 | 0/1 |
| 12 | 112975287 | A | C | rs7962920 | ALDH2 | 0.83 | 0.09 | SG | Alcohol metabolism | 1/1 | 1/1 | 1/1 | 1/1 | 1/1 | 0/0 | 0/0 | 0/0 | – | – | – |
| 15 | 28194482 | C | T | rs4778210 | OCA2 | 0.85 | 0.06 | SG | Pigmentation | 1/1 | – | 1/1 | 1/1 | 1/1 | 0/0 | 0/0 | 0/0 | 0/0 | 0/0 | 0/1 |
| 16 | 30602319 | G | A | rs59385041 | MAPK3, ZNF668 | 0.80 | 0.02 | SG | – | 1/1 | 1/1 | 1/1 | 1/1 | – | 0/0 | 0/0 | 0/0 | 0/0 | – | – |
| 16 | 89383725 | T | C | rs3114908 | MC1R | 0.75 | 0.63 | Tan | Pigmentation | 0/1 | 0/0 | 0/1 | 0/1 | 0/0 | 0/0 | 0/0 | 0/0 | 0/0 | 0/0 | 0/1 |
| X | 72050435 | G | A | rs201812199 | – | 0.84 | 0.01 | SG | – | 1/1 | 1/1 | – | – | 1/1 | 0/0 | – | – | 0/0 | 0/0 | 0/0 |

**Fig. 4 | Thirty-six risk loci where donor genotypes differ according to country.** Loci are either associated with keratinocyte cancer or tan response (Tan) or previously found to be under selection in Singaporean genomes; – for genotype indicates that no call could be made (Methods). Associated genes and mechanisms are suggested using phenome-wide association studies (PheWAS) and expression quantitative trait locus (eQTL) data from the Open Targets Genetics database. ECM, extra-cellular matrix; – indicates that the mechanism is unknown. Population allele frequencies are reported for East Asia (EA freq) and Great Britain (GBR freq) using data from the 1000 Genomes Project Phase 3 (Methods). Purple shading indicates the frequency of the alternative (ALT) allele in each population. Note that all loci in the *HYAL* region form part of the same linkage disequilibrium block on chromosome 3p.21. For each donor, green indicates homozygous reference (REF) alleles, yellow heterozygous and red homozygous ALT alleles.

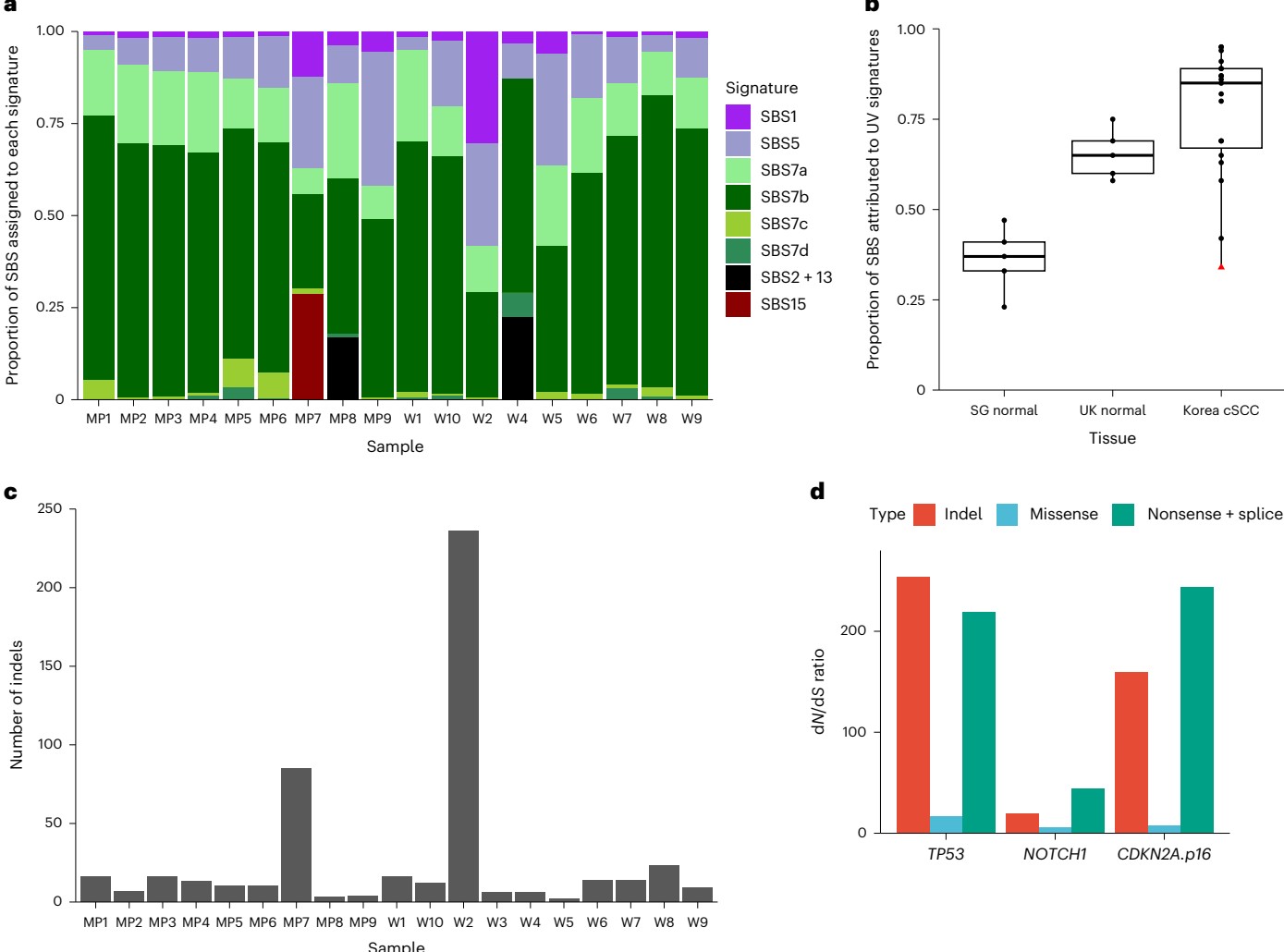

**Fig. 5 | Features of tumors from a low-incidence country (South Korea).**
**a**, Proportion of SBS attributed to reference mutational signatures according to donor (purple, aging signatures; green, UV radiation signatures; black, *APOBEC* mutagenesis; red, defective DNA mismatch repair). Korean samples are labeled as in the original study[12]. **b**, Proportion of substitutions attributable to UV radiation reference signatures SBS7a–d according to tissue (Singapore normal: $n = 5$ donors; UK normal: $n = 6$ donors; Korean cSCC: $n = 19$ tumors; red triangle, MP7,

a Korean tumor sample with defective DNA repair). Tukey box plot where the lower and upper hinges represent the first and third quartile, the center represents the median and the outliers are more than 1.5× the IQR. **c**, Number of insertions and deletions called in each Korean tumor sample. **d**, Ratio of observed and expected non-synonymous mutations for positively selected genes ($q < 0.01$ by dNdScv). Line drawn at $y = 1$.

CNA and increased selection of mutant *TP53*. This work supports previous studies suggesting that UV acts to promote carcinogenesis not only as a mutagen but also by promoting the expansion of preexisting *TP53* mutant clones[38], particularly of mutants that provide an advantage in UV-exposed skin, such as *TP53*^R248W (ref. 26).

There is evidence that aging UK epidermis is nearly saturated with competing mutant clones. It is possible that every cell carries a protein-altering mutation in a cancer-associated gene so that the growth of positively selected clones, such as *NOTCH1* mutants, is constrained. This shows how clonal dynamics can be altered by the environment. The comparatively stronger selection of *NOTCH1* and *NOTCH2* mutations in Singaporean skin compared to the UK and the underrepresentation of *NOTCH1* mutation in tumors is consistent with *NOTCH1* mutations providing a proliferative advantage but not increasing the risk of carcinogenesis[6,7]. In contrast, we do not find mutant *CDKN2A* to be under selection in aging epidermis, but it is common in cSCC, suggesting it drives carcinogenesis in keratinocytes.

In conclusion, comparing normal tissue across genetically distinct human populations that differ widely in cancer risk is most revealing.

In the high-incidence population, the normal somatic mutational landscape shares multiple features with the cancers that emerge from it, whereas in low-risk populations this is not the case.

## Online content

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

## Methods

### Sample collection

Sample collection and DNA sequencing of Singaporean skin samples was carried out as described for the UK samples[5]. Eyelid skin was collected from patients undergoing blepharoplasty surgery in Singapore. Informed consent was obtained in all cases under ethically approved protocols. The underlying fat and dermis were removed from the skin and the remaining tissue cut into approximately 0.25-cm$^2$ pieces. Each piece was incubated in 20 mmol l$^{-1}$ EDTA for 2 h at 37 °C. The epidermis was peeled from the dermis using fine forceps under a dissecting microscope and fixed for 30 min with 4% paraformaldehyde (PFA) (FD Neurotechnologies) before being washed three times in 1× PBS. The fixed epidermis was then cut into a contiguous array of approximately 40 samples per donor, each measuring 2 × 1 mm (Table 1). Donor ages are listed as ranges to help maintain donor anonymity. DNA was extracted from each sample using the QIAamp DNA Micro Kit (QIAGEN) by digesting overnight and according to the manufacturer's instructions. DNA was eluted using prewarmed AE buffer, where the first eluent was passed through the column twice more.

### DNA sequencing

Deep (approximately 700×), targeted sequencing was performed across 74 genes commonly mutated in cSCC and other cancers (Supplementary Table 1)[6]. This custom bait capture targets the exonic regions of these 74 genes, in addition to 1,734 SNPs across the genome to aid with copy number analysis. The targeted regions covered 0.67 Mb of the genome, with 0.33 Mb being exonic. Samples were multiplexed and sequenced on an HiSeq 2000 (Illumina) with version 4 chemistry to generate 75-bp paired-end reads. BAM files were mapped to the GRCh37d5 reference using the Burrows–Wheeler (BWA)-MEM[39] (v.0.7.17). Duplicate reads were marked using Biobambam2 v.2.0.86.

### Indel realignment and coverage

Reads around indels were realigned using GATK IndelRealigner and depth of coverage was calculated for targeted regions per sample using samtools (v.1.14) and bedtools (v.2.28.0). After removing duplicates and reads with a mapping quality of 25 or less and base quality of 30 or less, mean quality sequencing depth of coverage over all 428 epithelial samples (Singapore and UK) was calculated to be 749.0×.

### Copy number analysis

The allele frequency for each gene in a sample was estimated by statistically phasing heterozygous SNPs[6]. All samples of a donor were used as a panel to identify heterozygous SNPs at sites with at least 1,000× total coverage. Due to the variation in read depth across targeted regions, only copy number alterations which lead to an allelic imbalance, including LOH and gains, are detectable via this method (Supplementary Table 4).

### Mutation calling

Mutations were called using deepSNV (v.1.21.3)[40], a package designed to reliably detect mutations present in a small proportion of cells in a sample using a deeply sequenced panel of normal, sparsely mutated samples to determine a base-specific error model for each site in the targeted region. Mutations in each sample are then called by comparing the observed mutation rate against the background model using a likelihood ratio test[41]. Fifty-one samples of muscle or fat from UK donors, sequenced using the same method as the epithelial samples, were used to create a reference panel with a mean coverage of 42,611× over the targeted regions. This reference panel was used to call mutations across both Singapore and UK samples. Mutations were assumed to be germline and removed if present in more than 10% of all reads across all samples of a single donor. Across all samples, 13,850 mutations were detected, down to a minimum variant allele fraction of 0.0021 (median variant allele fraction = 0.015). Mean quality sequencing depth

of coverage exceeded 200× per sample in both Singapore and UK samples and the number of mutations detected did not correlate with sequencing depth in either group (Extended Data Fig. 3). Mutations were annotated using VAGrENT[42] (v.3.3.3).

### Spatial mapping of clones

Sampling the tissue in a grid of adjacent samples allows the mapping of large clones that spread over multiple samples. For all downstream analysis, identical mutations called in separate samples of the same donor were merged if the samples were known to have been within 10 mm of each other in the original tissue[41] (Supplementary Table 1). Mutations called in separate pieces of epidermis cut from the same individual were not merged because the distance between these samples cannot be accurately known. However, reanalysis with equally sized pieces of epidermis per donor confirmed that the number of samples per piece of epidermis does not confound estimates of mutation burden or clone size.

### Estimates of mutation burden and percentage mutant tissue

In the absence of CNA, the proportion of cells in a sample carrying a mutation can be estimated as double the variant allele fraction (the proportion of sequencing reads with a corresponding base change at that position). The genome regions targeted in this study cover genes commonly found to be mutated in cancers and consequently the mutation density observed is not likely to be representative of that genome-wide. Therefore, we used a method[41] to estimate the mutation burden per cell per megabase exclusively from synonymous sites in the bait region, excluding 32 samples where CNA was detected (Supplementary Tables 2 and 4). There was no evidence to suggest a difference in mean mutation burden between the eyebrows and eyelids of donors (two-sided Welch's $t$-test: $P = 0.14$). The percentage of mutant tissue upper bound estimates (Supplementary Tables 6 and 7) were calculated by summing the percentage of cells carrying at least one non-synonymous mutation of a sample, across all samples of a donor[41]. This assumes that, where possible, mutations occur in different cells of a sample. Lower-bound estimates assume that all mutations of a sample occur within the same cell. Patchwork plots were drawn by plotting the circular area of non-synonymous mutations from genes under selection after random selection of samples from each country, to make up 1-cm$^2$ tissue each[41]. A large $TP53$ mutant clone (that also carries a $NOTCH2$ mutation within it) spanned sixteen 2-mm$^2$ samples of skin in SG1. It is an outlier in terms of size compared to all other mutations (Extended Data Fig. 2b, Fig. 3 and Supplementary Table 1). The summed variant allele fraction (VAF) of this clone is 2.86, nearly four times larger than the next largest clone, a $NOTCH1$ mutant in a UK donor (summed VAF = 0.75). We report the percentages of $TP53$ mutant tissue according to country and the respective statistical significance both with and without this clone.

### Mutational signatures and selection

The trinucleotide context of each SBS was determined and the contribution of 49 reference mutational signatures (characterized across multiple cancers as part of the PCAWG study[25]) to this distribution was estimated using nonnegative matrix factorization with SigProfiler. To determine if mutational signature contribution differed according to country, we ran an unsupervised clustering algorithm. To compute significance, we used the pvclust R package, which uses bootstrap resampling techniques to compute a $P$ value for each hierarchical cluster. We found that the UK and Singapore were significantly distinct with respect to the signature contribution within each donor ($P < 1 \times 10^{-5}$). Low numbers of DBS and indels called precluded formal signature decomposition. Of the Korean cSCC dataset, two samples ('W-D_3' and 'W-D_4' in the original paper[12]) were excluded from the mutational signature analysis due to low mutation burden (fewer than 100 variants). Genes under selection were estimated using dNdScv[43] (Supplementary Table 5).

## Video model

We illustrated mutant clone growth in a sparsely (Supplementary Video 1) and densely (Supplementary Video 2) mutated environment to simulate clonal competition in skin from Singapore and the UK, respectively. In Supplementary Video 1, mutant clones of two arbitrary levels of fitness divide against a wild-type background of lower fitness until approximately 50% of the space is mutant. The starting mutation burden of Supplementary Video 2 is fourfold higher than for Supplementary Video 1, with cells dividing for the same number of divisions. Final mutant clone sizes are larger in a sparsely mutated environment (Supplementary Video 1) compared to a densely mutated environment (Supplementary Video 2), reflecting the different clone size distributions we observed according to country (Fig. 3f). The cell competition model used is a two-dimensional implementation of a Moran-like process[44].

## Donor genotyping

Reads across all samples of a donor were used to genotype donors (Supplementary Table 8). We genotyped donors using an SNP panel of 189 genomic sites associated with skin cancer risk, pigmentation and tan response to better explore the genetic differences between the UK and Singapore and gain more insight into cancer-protective mechanisms. Germline variants for each sample were called using GATK (v.4.3.0.0) best practices, with BAM files undergoing base quality recalibration before variant calling with HaplotypeCaller to produce a genomic variant call format (gVCF). Each of the gVCFs per sample was combined using GenomicsDBImport into a GenomicsDB database before joint calling using GenotypeGVCFs. We selected SNPs from the National Human Genome Research Institute-European Bioinformatics Institute GWAS and Open Targets databases and combined the loci associated with keratinocyte cancer (entry EFO_0010176–keratinocyte carcinoma), tan response (EFO_0004279–suntan) and 'Ease of skin tanning' (UK Biobank: 1727). We added these to 16 positively selected loci in Singaporean genomes. Associated genes and mechanisms are suggested for each SNP using PheWAS and eQTL data from the Open Targets Genetics database. Population allele frequencies are reported for East Asia and Great Britain using data from the 1000 Genomes Project Phase 3. We note that allele frequencies for Southeast Asia or Singapore are unavailable. East Asia allele frequencies were calculated from 504 individuals (approximately 300 Chinese, approximately 100 Japanese and approximately 100 Vietnamese). Great Britain allele frequencies were taken from 91 individuals across England and Scotland.

## Analysis of Korean cSCC exomes

Nineteen whole-exome sequenced Korean sample tumor-normal pairs were obtained from the SRA project SRP349018 using the SRA-toolkit (v.2.10.9). Sequence quality was assessed using FastQC (v.0.11.2) and visualized using MultiQC (v.1.13), confirming that mean Phred scores were all above 30 across 100 or 150 bp reads. The PCAP-Core workflow was followed to align the paired-end reads to the GRCh37d5 human reference genome using the BWA-MEM (v.0.7.17) with optical and PCR duplicates marked using Biobambam2 (v.2.0.86). Mean depth of coverage per sample was calculated using samtools (v.1.14) and bedtools (v.2.28.0) to be 61× (31–80×). Single-nucleotide variants were called using Caveman (v.1.17.4) with indels called using Pindel (v.3.3.0). Mutations were annotated using VAGrENT[42] (v.3.3.3). Genes under selection were estimated using dNdScv (v.0.0.1.0)[43] (Supplementary Table 5).

## TP53 codon bootstrapping

We applied nonparametric bootstrapping to estimate the significance of observing zero mutations at the R248 and R282 TP53 codons in Singaporean skin, given the decreased burden, decreased TP53 selection and decreased UV signature compared to UK skin. The nonparametric bootstrap statistical method is appropriate for these data because it does not make assumptions on distribution[45]. The method samples

from a given distribution (here, UK skin). For a bootstrap test, we simulated expected values for the mutation distribution in Singaporean skin given: (1) UK mutation distribution per donor and per sample of all, C>T/CC>TT, TP53 and R248W/R282W TP53 mutations and (2) the proportions of all, C>T/CC>TT and TP53 mutations between UK and Singapore. If the null hypothesis regarding the similarity of UK (adjusted) and Singapore distributions is true, we would expect that the simulated 'UK-adjusted' values would correspond to the Singapore observations, on average. We applied the 'rule of three' approach, often used in clinical trials[46], to estimate the robustness of zero outcomes in Singaporean skin. This approach suggests that the upper limit of a 95% confidence interval is $3/(n+1)$, where $n$ is the number of Singaporean samples. In this study, this value is $3/(191+1) = 0.0156$. We ran 1,000 simulations to obtain the estimation for an event of zero R248/R282 TP53 codon mutation counts in Singaporean samples. Of 1,000 simulations, none produced a value lower than 0.0156. The bootstrap simulation shows that (1) an expected mutation frequency at R248/R282 TP53 codons is around three mutations across all Singaporean samples and (2) the UK and Singaporean distributions are significantly different ($P < 0.001$).

## Statistics and reproducibility

No statistical method was used to predetermine sample size because the effect size was not known. No data were excluded from the analyses. The experiments were not randomized and the investigators were not blinded to allocation during the experiments and outcome assessment.

## Ethics and consent

Written informed consent was obtained in all cases. The study received ethical approval from the Nanyang Technological University Institutional Review Board, NHG study, protocol no. 2016/00659-AMD000l. The UK component of the study received ethical approval under UK approved protocols (research ethics committee references 15/EE/0152 NRES Committee East of England-Cambridge South and 15/EE/0218 NRES Committee East of England-Cambridge East).

## Reporting summary

Further information on research design is available in the Nature Portfolio Reporting Summary linked to this article.

## Data availability

Sequencing data have been deposited at the European Genome-phenome Archive with accession no. EGAD00001009666, titled 'Somatic mutations in facial skin from countries of contrasting skin cancer risk'. Downstream data are provided in the supplementary tables.

## Code availability

BAM files were mapped to the GRCh37d5 reference using the BWA-MEM[39] (v.0.7.17) and duplicate reads were marked using Biobambam2 v.2.0.86 (https://gitlab.com/german.tischler/biobambam2). Coverage was calculated using samtools (v.1.14) and bedtools (v.2.28.0). Mutations were called using deepSNV v.1.21.3 (https://github.com/gerstung-lab/deepSNV)[40,41] and annotated with VAGrENT[42] v.3.3.3. All downstream analyses were conducted in R v.4.1.3 with data visualization and statistical analysis conducted using the following R packages: Biostrings, car, deepSNV, dNdScv, GenomicRanges, ggrepel, ggpubr, igraph, MASS, plotrix, pvclust (https://CRAN.R-project.org/package=pvclust), Rsamtools, rstatix, seqinr and tidyverse. Mutational signature analysis was conducted using SigProfiler v.1.1.13. Germline variants were called using GATK v.4.3.0.0 best practices. The Korean cSCC samples were obtained using the SRA-toolkit v.2.10.9, assessed using FastQC v.0.11.2 and MultiQC v.1.13 and aligned to GRCh37d5 using the BWA-MEM v.0.7.17 (https://github.com/lh3/bwa) and Biobambam2. Mutations were called using Caveman v.1.17.4 and Pindel v.3.3.0, and annotated using VAGrENT[42]. The modeling code is available at https://github.com/irinaabnizova/cell_competition_2D.

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

## Acknowledgements

This work was supported by grants from the Wellcome Trust to the Wellcome Sanger Institute (grant nos. 098051 and 296194). P.H.J. is supported by a Cancer Research UK Programme Grant (grant no. C609/A27326). B.A.H. acknowledges support from the Royal Society (grant no. UF130039).

## Author contributions

P.H.J. and E.B.L. designed the study. S.M.Y., I.S., M.T., J.H. and J.C.F. performed the experiments. C.K., M.W.J.H. and R.K.S. analyzed the sequencing data. I.A. performed the statistical analyses and clone simulations, and was supervised by B.A.H. P.H.J. supervised the research. C.K. and P.H.J. wrote the paper; all authors commented on the paper.

## Competing interests

The authors declare no competing interests.

## Additional information

**Extended data** is available for this paper at https://doi.org/10.1038/s41588-023-01468-x.

**Correspondence and requests for materials** should be addressed to Philip H. Jones.

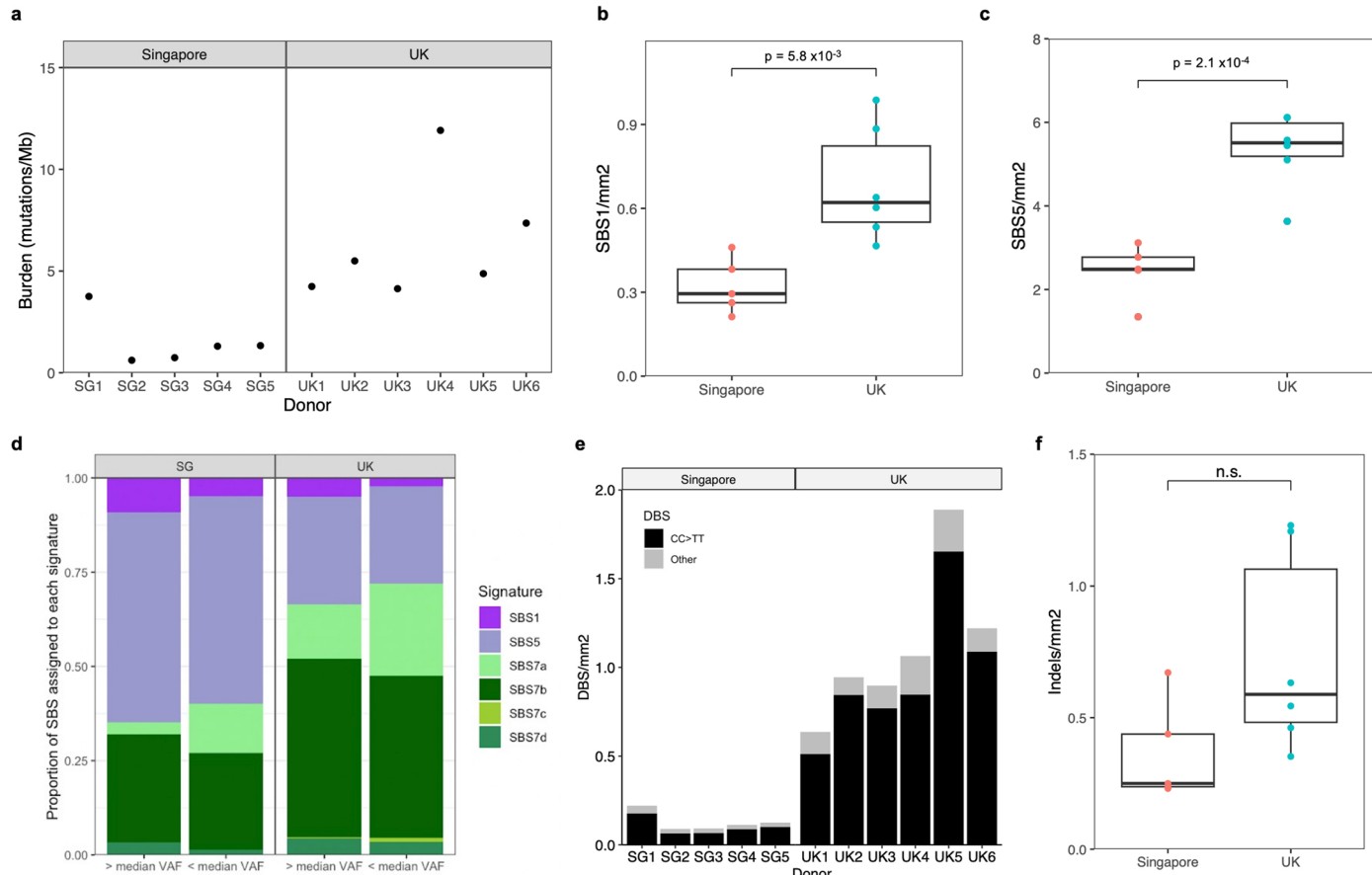

**Extended Data Fig. 1 | Mutational burden and signatures. a** Estimates of genome-wide mutation burden per donor. **b** Proportion of SBS1 mutations/mm² per donor (n = 11) by country (two-sided t-test: p = 5.8 × 10⁻³). Tukey boxplot where lower and upper hinges = 1ˢᵗ and 3ʳᵈ quartile, centre = median, outliers > 1.5 x inter-quartile range. **c** Proportion of SBS5 mutations/mm² per donor (n = 11) by country (two-sided t-test: p = 2.1 × 10⁻⁴). Tukey boxplot where lower and upper hinges = 1ˢᵗ and 3ʳᵈ quartile, centre = median, outliers > 1.5 × inter-quartile range. **d** Proportion of SBS assigned to each signature, split by mutations above

and below median variant allele frequency (VAF) for each country (Pearson's chi-square UK: p < 2.2 × 10⁻¹⁶; SG: p = 1.3 × 10⁻¹⁴). **e** Counts of double-base substitutions (DBS) per mm² of skin of donors from each country (UK mean = 1.08 DBS/mm², SG mean = 0.126 DBS/mm², two-sided Welch's t-test: p = 2.3 × 10⁻³). **f** Count of insertions and deletions per mm² in each donor (n = 11) by country (UK mean = 0.72 indels/mm², SG mean = 0.37 indels/mm², two-sided Welch's t-test: p = 0.07). Tukey boxplot where lower and upper hinges = 1ˢᵗ and 3ʳᵈ quartile, centre = median, outliers > 1.5 x inter-quartile range.

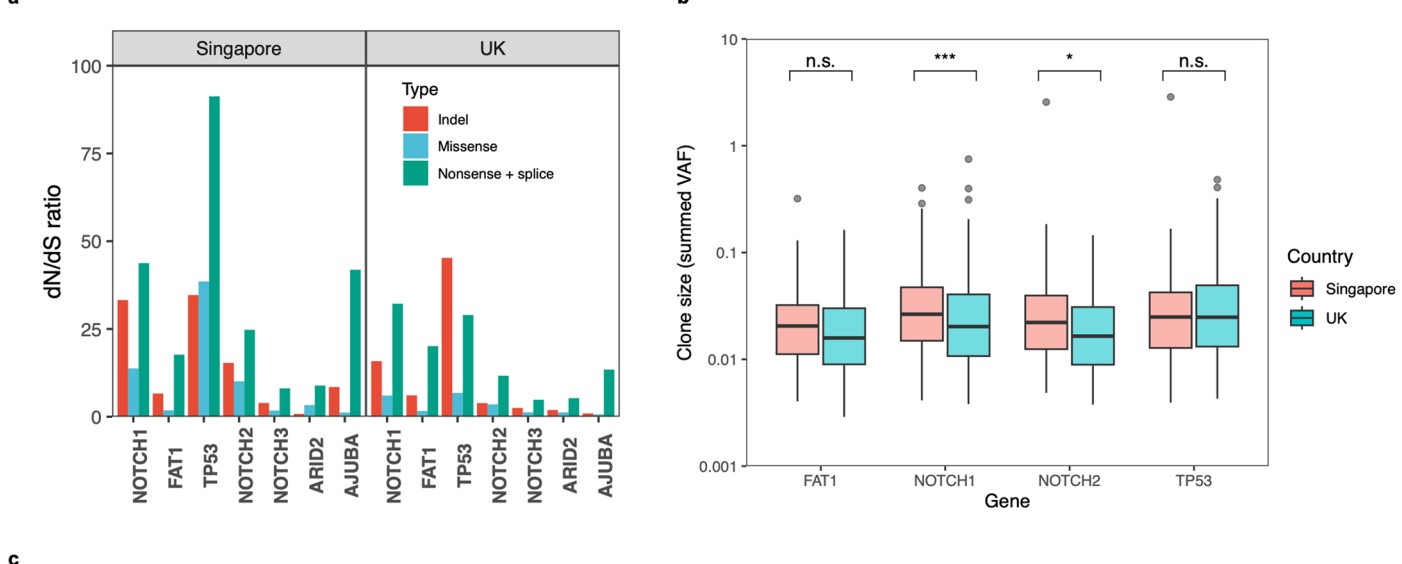

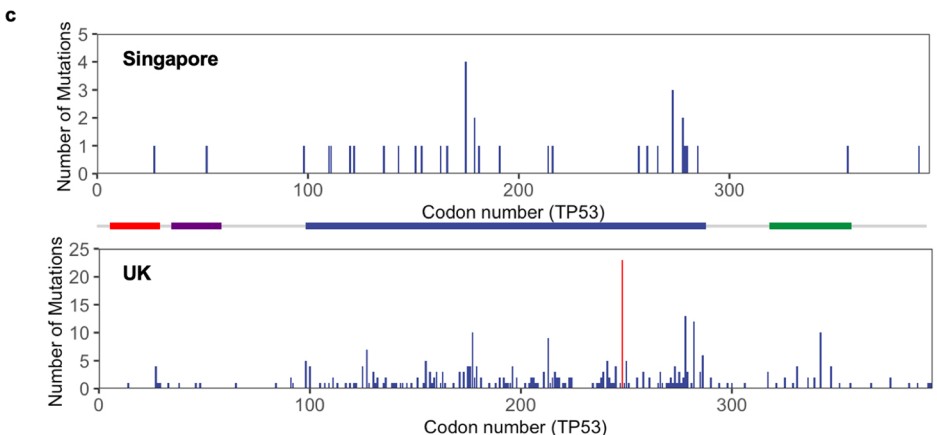

**Extended Data Fig. 2 | Mutant clone selection differs by country. a** Ratio of observed/expected non-synonymous mutations for positively selected genes by country (q < 0.01). No synonymous mutations were detected in Singaporean skin for *TP53* and *AJUBA*, leading to high $d$N/$d$S ratios. Line drawn at y = 1. **b** Sizes of all clones with protein-altering mutations in the top four positively selected genes, by country (samples with known CNA removed), two-sided t-test adjusted by Bonferroni multiple test correction: p = 6 × 10⁻⁴ (*NOTCH1*, n = 1,020), 1.5 × 10⁻² (*NOTCH2*, n = 390), 0.68 (*FAT1*, n = 432) and 1.0 (*TP53*, n = 366). Tukey boxplot where lower and upper hinges = 1$^{st}$ and 3$^{rd}$ quartile, centre = median, outliers > 1.5 x inter-quartile range. **c** Distributions of mutations across codons in *TP53*. The most frequently mutated codon in cancer, R248 (shown red), is the most common codon change in UK skin but is absent in Singaporean skin.

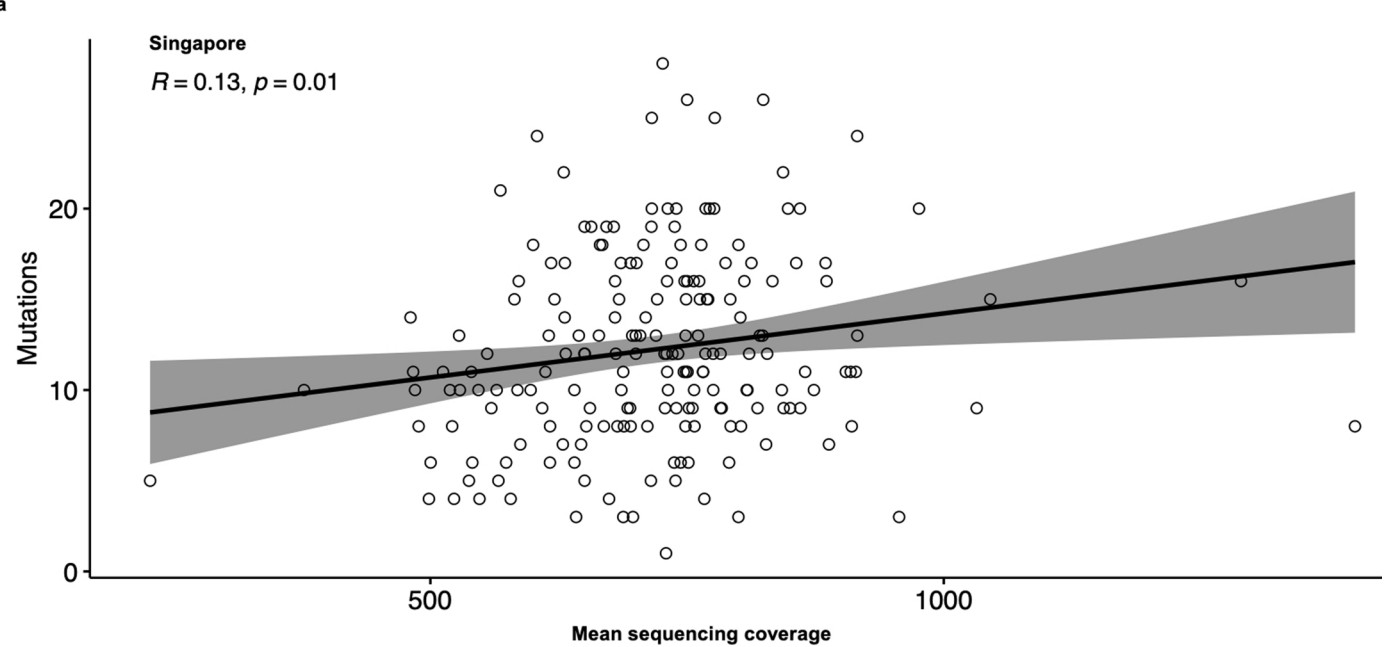

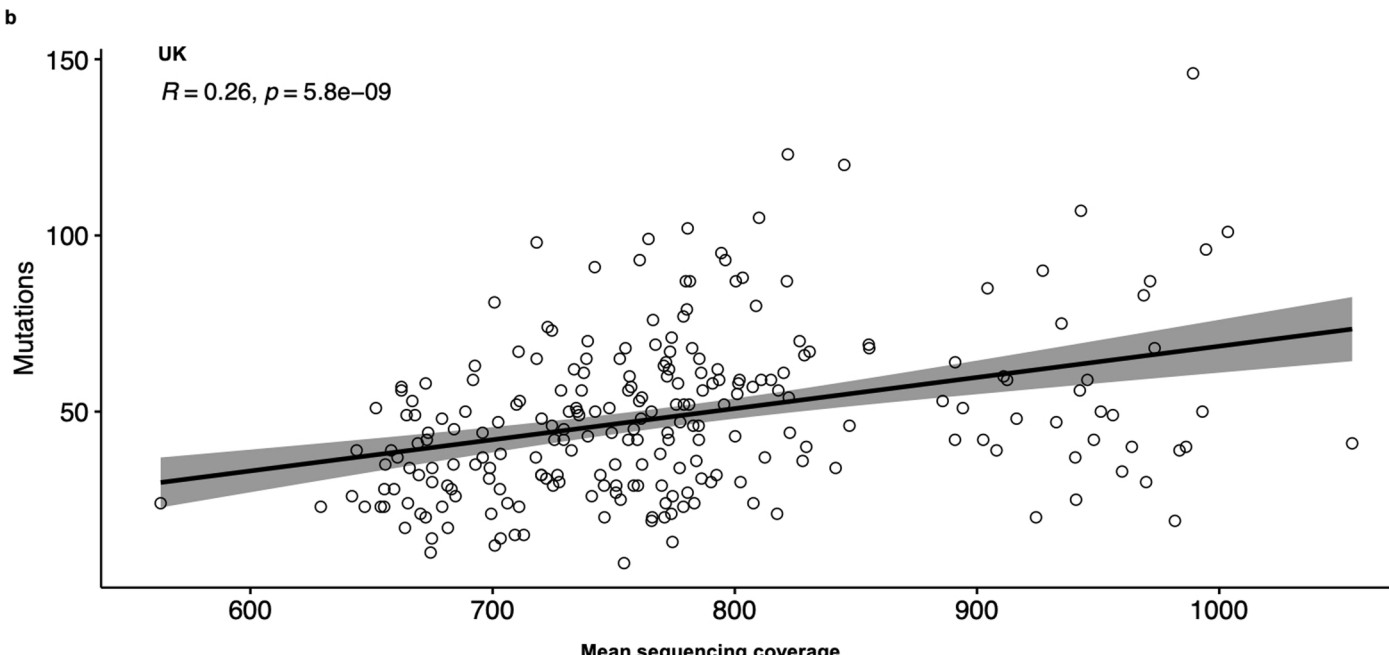

**Extended Data Fig. 3 | Sample mutation counts do not correlate with sequencing coverage.** Correlation between mean quality sequencing coverage per sample and the number of mutations detected for **a** Singapore and **b** UK (error bands = 95% confidence interval). Mean depth of coverage was calculated after removing off-target reads, duplicates and those with mapping quality of 25 or less and base quality of 30 or less. Samples of neither country show a correlation between depth of coverage and mutation counts (linear regression: (SG) R = 0.13, p = 0.01; (UK) R = 0.26, p = $5.8 \times 10^{-9}$).

|---|---|

# Reporting Summary

Please do not complete any field with "not applicable" or n/a. Refer to the help text for what text to use if an item is not relevant to your study.
For final submission: please carefully check your responses for accuracy; you will not be able to make changes later.

## Statistics

For all statistical analyses, confirm that the following items are present in the figure legend, table legend, main text, or Methods section.

| n/a | Confirmed | |
|---|---|---|
| ☐ | ☒ | The exact sample size (*n*) for each experimental group/condition, given as a discrete number and unit of measurement |
| ☒ | ☐ | A statement on whether measurements were taken from distinct samples or whether the same sample was measured repeatedly |
| ☐ | ☒ | The statistical test(s) used AND whether they are one- or two-sided *Only common tests should be described solely by name; describe more complex techniques in the Methods section.* |
| ☒ | ☐ | A description of all covariates tested |
| ☐ | ☒ | A description of any assumptions or corrections, such as tests of normality and adjustment for multiple comparisons |
| ☐ | ☒ | A full description of the statistical parameters including central tendency (e.g. means) or other basic estimates (e.g. regression coefficient) AND variation (e.g. standard deviation) or associated estimates of uncertainty (e.g. confidence intervals) |
| ☐ | ☒ | For null hypothesis testing, the test statistic (e.g. *F*, *t*, *r*) with confidence intervals, effect sizes, degrees of freedom and *P* value noted *Give P values as exact values whenever suitable.* |
| ☒ | ☐ | For Bayesian analysis, information on the choice of priors and Markov chain Monte Carlo settings |
| ☐ | ☒ | For hierarchical and complex designs, identification of the appropriate level for tests and full reporting of outcomes |
| ☐ | ☒ | Estimates of effect sizes (e.g. Cohen's *d*, Pearson's *r*), indicating how they were calculated |

*Our web collection on statistics for biologists contains articles on many of the points above.*

## Software and code

Policy information about availability of computer code

| Data collection | No software was used to collect data |
|---|---|
| Data analysis | BAM files were mapped to the GRCh37d5 reference using BWA-MEM (v0.7.17) and duplicate reads marked using Biobambam2 (https://gitlab.com/german.tischler/biobambam2, v2.0.86). Coverage was calculated using samtools (v1.14) and bedtools (v2.28.0). Mutations were called using deepSNV (https://github.com/gerstung-lab/deepSNV, v1.21.3) and annotated using VAGrENT (v3.3.3). All downstream analysis was conducted in R (v4.1.3) with data visualisation and statistical analysis conducted using R packages: Biostrings, car, deepSNV, dndscv (v0.0.1.0), GenomicRanges, ggrepel, ggpubr, igraph, MASS, plotrix, pvclust, Rsamtools, rstatix, seqinr and tidyverse. Mutational signature analysis was conducted using SigProfiler (v1.1.13). Germline variants were called using GATK (v4.3.0.0) best practices. Korean cSCC samples were obtained using SRA-toolkit (v2.10.9), assessed using FastQC (v0.11.2) and MultiQC (v1.13) and aligned to GRCh37d5 using BWA-MEM (v0.7.17, https://github.com/lh3/bwa) and Biobambam2 (v2.0.86, https://gitlab.com/german.tischler/biobambam2). Mutations were called using Caveman (v1.17.4) and Pindel (v3.3.0) and annotated using VAGrENT (v3.3.3). All modelling code is available at https://github.com/irinaabnizova/cell_competition_2D |

For manuscripts utilizing custom algorithms or software that are central to the research but not yet described in published literature, software must be made available to editors and reviewers. We strongly encourage code deposition in a community repository (e.g. GitHub). See the Nature Portfolio guidelines for submitting code & software for further information.

## Data

Policy information about availability of data

All manuscripts must include a data availability statement. This statement should provide the following information, where applicable:
- Accession codes, unique identifiers, or web links for publicly available datasets
- A description of any restrictions on data availability
- For clinical datasets or third party data, please ensure that the statement adheres to our policy

Sequencing data has been deposited at EGA and is accessible with the accession number EGAD00001009666, title: Somatic mutations in facial skin from countries of contrasting skin cancer risk.

## Research involving human participants, their data, or biological material

Policy information about studies with human participants or human data. See also policy information about sex, gender (identity/presentation), and sexual orientation and race, ethnicity and racism.

| | |
|---|---|
| Reporting on sex and gender | Male and female sex is reported and both are represented in the data. Males: n = 6, females: n = 5 |
| Reporting on race, ethnicity, or other socially relevant groupings | Human donors were categorized by the country in which they live (Singapore and the UK). We then genotyped donors for relevant SNPs and showed donors were representative of the wider populations of each country. |
| Population characteristics | Donors were aged 28-79 years (Singapore mean = 62 years, UK mean = 68 years), with both males/females, smokers/non-smokers and indoor/outdoor workers from each country. |
| Recruitment | Participants were undergoing blepharoplasty or browplexy surgery in Singapore or the UK. This is performed for age-related loss of elasticity of the dermis. No other selection criteria were applied. Participants did not receive compensation and we are not aware of any recruitment bias that may impact the results. |
| Ethics oversight | Written informed consent was obtained in all cases. The study received ethical approval from the Nanyang Technological University Institutional Review Board (www.ntu.edu.sg/research/research-integrity-office/institutional-review-board/the-ntu-institutional-review-board-(irb), protocol number, NHG study 2016/00659-AMD000l). The UK component of the study received ethical approval under UK approved protocols (Research Ethics Committee references 15/EE/0152 NRES Committee East of England-Cambridge South and 15/EE/0218 NRES Committee East of England-Cambridge East). |

Note that full information on the approval of the study protocol must also be provided in the manuscript.

# Field-specific reporting

Please select the one below that is the best fit for your research. If you are not sure, read the appropriate sections before making your selection.

☒ Life sciences          ☐ Behavioural & social sciences          ☐ Ecological, evolutionary & environmental sciences

# Life sciences study design

All studies must disclose on these points even when the disclosure is negative.

| | |
|---|---|
| Sample size | Sample size was estimated from previous studies (PMID: 25999502, PMID: 33087317). |
| Data exclusions | No data were excluded from the study. |
| Replication | Multiple donors were sampled from each country, with a large effect size between countries. The scarcity of donated healthy human tissue makes the experiment difficult to replicate but the high consistency with previously published work gives us confidence on the reproducibility. |
| Randomization | Consecutive consenting eligible donors were recruited into the study without other selection. Allocation into experimental groups was based on the country in which the participant lives (our independent variable) and therefore randomization is not relevant to this study. |
| Blinding | Blinding is not relevant to data collection or data analysis in this study as DNA sequencing, mutation calling and downstream algorithms are not affected by investigator knowledge of a donor's country. |

# Reporting for specific materials, systems and methods

We require information from authors about some types of materials, experimental systems and methods used in many studies. Here, indicate whether each material, system or method listed is relevant to your study. If you are not sure if a list item applies to your research, read the appropriate section before selecting a response.

<div style="display:flex">
<div>

## Materials & experimental systems

| n/a | Involved in the study |
|-----|------------------------|
| ☒ ☐ | Antibodies |
| ☒ ☐ | Eukaryotic cell lines |
| ☒ ☐ | Palaeontology and archaeology |
| ☒ ☐ | Animals and other organisms |
| ☒ ☐ | Clinical data |
| ☒ ☐ | Dual use research of concern |
| ☒ ☐ | Plants |

</div>
<div>

## Methods

| n/a | Involved in the study |
|-----|------------------------|
| ☒ ☐ | ChIP-seq |
| ☒ ☐ | Flow cytometry |
| ☒ ☐ | MRI-based neuroimaging |

</div>
</div>

nature portfolio | reporting summary

April 2023

