## [Peer Review File · Nature Genetics]

Peer Review Information

Manuscript Title: Somatic mutations in facial skin from countries of contrasting skin cancer risk

Corresponding author name(s): Professor Philip (H) Jones

Reviewer Comments & Decisions:

Decision Letter, initial version:

4th Jan 2023

Dear Professor Jones,

Your Letter, "Somatic mutations in facial skin from countries of contrasting skin cancer risk" has now been seen by 3 referees. You will see from their comments below that while they find your work of interest, some important points are raised. We are interested in the possibility of publishing your study in Nature Genetics, but would like to consider your response to these concerns in the form of a revised manuscript before we make a final decision on publication.

To guide the scope of the revisions, the editors discuss the referee reports in detail within the team, including with the chief editor, with a view to identifying key priorities that should be addressed in revision and sometimes overruling referee requests that are deemed beyond the scope of the current study. In this case, we'd ask that you address all the comments, paying particular attention to those that are focused on deriving mechanistic (rather than lifestyle) underpinnings of your findings.

Please do not hesitate to get in touch if you would like to discuss these issues further.

We therefore invite you to revise your manuscript taking into account all reviewer and editor comments. Please highlight all changes in the manuscript text file. At this stage we will need you to upload a copy of the manuscript in MS Word .docx or similar editable format.

*2) If you have not done so already please begin to revise your manuscript so that it conforms to our Letter format instructions, available [here](http://www.nature.com/ng/authors/article_types/index.html). Refer also to any guidelines provided in this letter.

[redacted]

We hope to receive your revised manuscript within four to eight weeks. If you cannot send it within this time, please let us know.

Sincerely,

Safia Danovi
Editor
Nature Genetics

Referee expertise:

Referee #1: skin cancer risk, genetics

Referee #2: skin cancer and ageing

Referee #3: genomics, cell competition (signed report)

Reviewers' Comments:

Reviewer #1:

Remarks to the Author:

Summary

The authors have systematically compared the mutational landscape of a large number of biopsies (191) taken from a sun-exposed site from 5 individuals from Singapore. They have contrasted these results with a similar dataset from 6 people from the UK.

The samples from individuals from Singapore showed much lower rates of mutations and smaller/fewer mutant clones. The overall mutational burden was also very low compared to the samples from the UK, and proportionally lower was the signal from UV-induced mutations even though Singapore experience 2-3 fold higher rates of exposure.

WES of cSCC lesions from 19 Korean individuals (chosen as were similar to the Singapore population) was very similar to the mutation profiles of SCCs from elsewhere in the world suggesting that while the profiles of healthy skin may differ by population and region, SCC tumours follow a similar oncogenic process across populations.

The discussion of the reasons for these differences is cursory. The clearest difference, and most likely cause, is that the 6 individuals from Singapore have a greater Fitzpatrick score (3-4) vs the UK samples (2), and essentially lack any low pigmentation risk alleles (Figure 4) indicating they likely have a greater degree of skin pigmentation and tanning ability than UK samples. This is of interest both to better understand the causes of differences in ageing skin, but also to highlight that better adherence to UV protective behaviours in UK populations (and other fair-skinned populations) may reduce the mutation burden in the skin.

Comments

1. Abstract line one, and final sentence page 2. References 1-3, used to report the relative incidences of SCC across Singapore and the UK are from 2005 and earlier - are there no more recent surveys of skin cancer incidence in these countries? Figure 1a shows incidence rates up until 2015 suggesting different sources have been used then were cited (ref 1) in figure 1's footnote;

Suggested action: check/update references

2. The final sentence of the abstract offers no conclusion as to why skin from a low-risk country does not show multiple features convergent with cancer; the likely answer from the data is the degree of skin pigmentation and propensity to tan. This should be addressed. Also, see points 4 and 11.

Suggested action: Update abstract/text to address

3. Introduction, paragraph 1, page 2, sentence 5: Refs 18-21 aren't citations to a large twin study (I assume the authors mean Mucci 2016 PMID: 26746459) nor are they all citations to GWAS of KC cancer. 18 is a PRS study, 20 isn't a GWAS, and there are other published KC GWAS other than refs 19 and 21 that should be cited.

Suggested action: check/update references

4. Final sentence of introduction paragraph 1: In my opinion saying that some GWAS SNPs for KC are linked to skin pigmentation somewhat undersells this fact. For example, in ref 21 the majority of the most strongly associated loci (and thus represent the largest combination of effect sizes and minor allele frequency, and thus the proportion of trait variance explained) are all pigmentation SNPs. While other pathways are important in KC risk, pigmentation is by and far the strongest genetic risk for KC, which is why the rates are so elevated in the UK and other countries with majority fair-skinned populations. Related - in the introduction paragraph 3 while there may be additional genes under selection in East Asian populations that are relevant to skin cancer risk, a likely major reason for the reduced (genetic) risk for KC is the absence or greatly reduced frequency of low pigmentation risk alleles enriched in European ancestry populations e.g.

- rs12203592, a pigmentation SNP influencing IRF4 function and a KC risk SNP has a low pigmentation T allele freq of ~14% in EUR pops, and < 0.01% in East Asian populations.
- rs1805007, one of the strongest red hair alleles and a strong risk allele for KC, has a risk T allele freq of ~7% in EUR but only 0.07% in East Asian populations (source: GNOMAD).

Suggested action: The importance of pigmentation SNPs/genes should be more fully and clearly discussed since it has bearing on their later results.

5. Introduction page 3 - sentence 1. Expand briefly on the behaviour differences with respect to UV exposure and if relevant discuss how it may impact the differences in mutational burden observed.

Suggested action: Update abstract/text to address

6. Introduction page 3 - sentence 2. OCA2 isn't a precursor to melanin; it is a channel transporter protein most likely involved in maintaining the pH of melanosomes, allowing Tyrosinase's production of melanin - see PMID 32333855

Suggested action: Update abstract/text to address

7. Table 1

- a. It would be very helpful to merge the key mutational load columns from Sup table 12A into this table
- b. Why are the age listed as ranges e.g SG1 is 68-71?
- c. While the mean ages are roughly similar between the UK and SG samples, SG2 is young (28-31) which is relevant given the premise here is to report on ageing skin. The similarity of this sample to the others in terms of mutation load etc should be discussed - e.g. are they an outlier?
- d. The most striking difference in this table is that all the SG samples have higher Fitzpatrick skin types. This makes it difficult to interpret what differences are due to country (and UV exposure) vs host factors (a more photoprotective skin type) as they are confounded. The authors should discuss this and report on the regression of Fitzpatrick score onto mutational load - how much of the variation in mutational load is explicable by Fitzpatrick score?

Suggested action: update tables to address (a), and clarify table or footnote to address (b). (c) and (d) please update analysis/manuscript as required

8. Methods - multiple testing. There are a large number of tests performed - multiple testing correction should be applied and the impact in terms of the significance of results reported.

Suggested action: List and report total number of tests and adjust the manuscript appropriately.

9. Page 8 final paragraph; the justification for removing the large TP53 clone from SG1 is not clearly presented. Why is it valid to exclude this sample? This should be made clear in the methods/results

Suggested action: Discuss/address in manuscript

10. Page 10, second last sentence. "-we would still expect to observe at least 3 mutations at codons R248/R282 across all Singaporean samples" isn't clear. If I understand it correctly the authors are suggesting 0 observations out of 191 samples is unusual given they expected 3 out of 191 - is that especially unlikely given the events are rare? If that is the correct interpretation is this difference significant using an appropriate non-parametric test? If not clarify this result.

Suggested action: Clarify in manuscript and include additional results if relevant

11. Figure 4 indicates that the SG population essentially lacks any of the major pigmentation risk alleles. This is interesting (but perhaps not surprising). In general the implications of this aren't really addressed in the manuscript, and relates to point 4 - that the likely main reason the SG samples have lower mutation burdens than UK samples is they have greater skin pigmentation and that greater protection against UV damage. Specific comments about the SNP selection and figure 4:

- a) Please include chr and position and present SNPs in their chr/bp order order
- b) Please include an appropriate reference population allele freqs for EUR and EA (e.g. from GNOMAD) as a column to help contextualise the results. This is relevant because many SNPs are rare (e.g. rs192481803 has a MAF ~ 0 in all pops which is likely why everyone is missing the risk allele) and/or many of the risk alleles related to pigmentation are essentially absent in East and South Asian populations e.g. see point 4.
- c). Please check if the listed SNPs are in LD with each other at least in EUR pops (as their BCC/SCC associtons were identified n EUR ancestry pops) e.g. rs4268748 is an intronic variant near MC1R, and is in LD with the rs1805007 functional missense allele (LD r^2 0.23 in EUR) which is likely why it has previously been reported for BCC/SCC. As rs1805007 is wildtype in SG samples I am not sure it is relevant this SNP is heterozygous in some SG samples. The authors should consider removing SNPs in LD with other functional SNPs if they exist.
- d). Some SNP IDs are out of date e.g. the ID rs74664507 has been merged with rs11445081
- e). Some listed SNPs have known functions/associations assigned since the publication of the cited reviews (41 and 42) e.g. rs74664507 / rs11445081 is strongly associated with skin colour. The authors should survey more recent literature, or databases like Open targets genetics (e.g. https://genetics.opentargets.org/variant/9_16913838_T_A) to update the figure to better reflect current information.

Suggested action: update/check SNP data entered into figure 4 and related to points 2 and 4 address implications in the manuscript.

12. Methods - page 17. Please provide more details on the reference panel used to call mutations. Specifically, what was the ancestry of the reference population used, and did it match each sequenced population e.g. using an off-ancestry reference panel could be expected to increase the mutation detection rate due to real SNP differences between panel and target.

Suggested action: update methods and discuss if relevant.

13. Methods page 18 paragraph 1 final sentence + sup figure 3. Please report the significance value for the correlation $R = 0.33$ between coverage and mutations detected.

Suggested action: Report significance, and if this is significant the statement starting “no evidence” will need to be adjusted and this limitation addressed in the discussion

14. Methods video model + model images. Are these simulations derived in any way from the data other than starting the mutation burden 4 fold higher in video 2? E.g. were the model’s settings for fitness derived from the reported dataset? If so please state so clearly. If not, please clarify what these videos are showing us about the data presented, or how they are otherwise providing nuance to the manuscript.

Suggested action: Clarify methods for these videos and their relevance to the paper

Reviewer #2:

Remarks to the Author:

This paper compares the accumulated mutations in human epidermis of individuals of two different ethnic backgrounds, UK and Singapore. The two ethnicities have a different lifetime risk of developing cSCC, independently of UV exposure. The comparison shows individuals from the high-risk ethnicity (UK) have a distinct pattern of mutations and mutational processes that overlap with cancer-driving cSCC genes, and individuals from the cSCC low risk group (Sing) in contrast have fewer mutations, fewer mutations in known SCC driver genes, and importantly, have an increase in the mutations that are linked to skin ageing but not skin cancer.

This study is important, novel and opens many new questions that will impact scientists and clinicians working in the life sciences, not only cancer biology. The potential impact to medicine is clear.

The main limitation is there are no valid hypotheses put forward to explain the difference in mutational landscape between UV exposed skin of different ethnic backgrounds, no experimental work to explain why different ethnic backgrounds develop/select different mutations.

Questions

a. Previous work shows sbs1 sig reflects the number of cell divisions, as it increases at a constant rate per cell division. The rate of cell division in human skin depends on sun exposure and age. Can the authors infer whether the rate of sbs1 mutations over time is equal in both skin types, taking into account sbs7? It is interesting to test whether cell cycle regulation underpins ethnic differences.

b. If there are differences in the cell division rate inferred from the sbs1 rate, adjusted by age and sbs7, for example faster cycling in UK skin, could these differences link to less time for DNA repair, i.e more mutations? Have they compared DNA repair?

c. You propose that notch1 can prevail and expand in Singapore skin due to less competition from other clones. You propose this is a possible explanation for the difference in clone selection. This possibility could be confirmed in skin of young UK people, with fewer mutations. Does younger UK skin have high notch1, and will ageing reduce these in favour of the more UK-prevalent p53 clones? My hypothesis is that this will not be the case, and young skin will select R348W and R282W TP53. There are no alternative hypotheses in the paper to explain why the accumulation of specific mutations is not stochastic across both skins.

d. Is there any evidence ethnic loci risk scc variants promote selection of specific DNA mutations? Can this be tested?

e. Many of the high risk loci are linked to pigmentation. Is the rationale that Singapore skin will make "more" pigment to protect better from UV? In reality, Singapore skin has a similar level of pigmentation compared to Uk skin, at least the difference is not large enough to explain the dramatic difference in incidence. Moreover, many Mediterranean countries have extremely high incidence of scc, compared to Singapore, and these differences cannot be explained by pigmentation level alone. Do they have any mechanistic insight to what any of the variants do?

Reviewer #3:

Remarks to the Author:

We know that skin cancer risk varies enormously across populations around the world. But the question is – why? King and colleagues present an analysis of the somatic mutational landscapes of eyelid skin from subjects from Singapore (~17X lower skin cancer risk compared to UK, despite much higher UV exposure), comparing results from a previously reported UK cohort. The use of eyelid samples should control for differences in skin coverage with clothing, with the caveat that sunglasses can block UV exposure. Their studies reveal striking differences in mutational burden and signatures, selection, and mutational hotspots in expansions across the two cohorts, with the UK skin exhibiting greater features found in skin cancers. Results are very clearly presented and well written. Data are supportive of claims, which are not overstated. Methods are well described. One caveat of the study is that they are comparing two cohorts that differ both by genetics and by sun/UV exposure, making it more difficult to ascertain relative contributions to the mutational patterns observed. Still, it is interesting that mutational burden and “pro-cancer” selective patterns are greater for the UK subjects despite substantially less overall UV exposure, perhaps relating to greater genetic risk (resulting in less protection by skin pigmentation).

In all, this study makes an important contribution to the field of somatic mutations in normal tissues and how they could contribute to cancer risks, including risks that vary across populations. Major findings include:

1. Mean mutational burden is 4X higher in UK subjects than Singapore subjects, and the UK subjects had substantially greater UV mutational signatures and 8X as many double base substitutions characteristic of UV exposure. CNAs were evident in 13% of UK eyelid specimen compared to just 1.0% of Singaporean. Thus, mutational burden correlates with skin cancer risk.
2. Perhaps one of the most interesting findings from this study is that fitness landscapes appear to differ for the two populations, with NOTCH1 and NOTCH2 clonal expansions occupying less and TP53 expansions occupying more of the epidermis for the UK subjects than expected by the mutational burden. For the TP53 expansions, this was only true if an outlier clone was removed from one Singaporean subject (a reasonable exercise). In all, while about half of the epidermis in the Singapore group is occupied by clones with disruptive mutations in at least one of the 74 examined genes, this number is about 95% for the UK cohort. The distribution of clone size is smaller for the UK cohort, perhaps indicating clonal competition, which may also explain why NOTCH1/2 clones are significantly smaller (suppressed) in the UK skin samples (in contrast, TP53 clone size was similar across the cohorts). This competition is illustrated in 2 videos of a computational model.
3. They observed differences in the mutational spectra for TP53, FGFR3 and RAS genes, with the mutations most commonly observed in skin cancers (or cancers in general) almost exclusively restricted to the UK cohort.

4. Not surprisingly, genetic (germline) variants associated with pigmentation and SCC risk were differentially found in the UK cohort. Other risk alleles showed similar distributions across the two cohorts.

5. Analyses of cSCC from South Korea (where risk of SCC is similar to that of Singapore) revealed that the UV mutational signature still dominates. If we can extrapolate to SCCs from Singapore (not ideal, but not totally unreasonable), this suggests that despite lower mutational burden in the skin and a reduced UV signature in Southeast Asians, that the cancers are still driven by UV mutagenesis.

In all, their data support their conclusion that UK skin shares more features with skin cancers (in terms of mutational burden and signatures, CNAs, genes under selection, and hotspot mutations) compared to Singaporean skin.

Questions to address:

1. Do Brits tend to holiday in warmer places with more sun exposure, and perhaps experience sun burns? Periodic high exposure may be worse than regular moderate exposure, as the skin can adapt to the latter. Could increased vacationing of Brits closer to the equator have contributed to the 2-3X rise in skin cancer incidence over the last few decades?

2. Were the UK donors of North European descent (Anglo-Saxon) and the Singapore donors of South Asian descent? It's important to make this clear, as there has clearly been a lot of migration of humans.

3. From Supplementary table 3 it can be observed that the number of non-UV associated SBS1 and SBS5 mutations are more than twice in the UK cohort compared to the Singapore cohort – this should be pointed out in the manuscript. One interpretation of this observation is that the skin of UK subjects is more prone to mutation-driven clonal expansions independent of UV mediated mutagenesis, such as UV induced selection; alternatively, these mutations could be due to indirect UV mutagenesis or other exposures.

4. For Fig S2a, there seems to be a substantial difference in dN/dS for missense and nonsense+splice TP53 mutations comparing Singapore and UK subjects, both of which are much higher in the Singapore subjects. Could this just reflect the overall lower mutational burden in the Singapore cohort, and thus when a TP53 mutation is detected it is more likely to alter the protein? Some speculation might be warranted, unless we are just missing something.

5. Also, with regard to TP53, while it's reasonable to assume that the large TP53 clone that is observed in donor SG1 is an outlier, it still begs the question if it might have some biological significance. The mutation in question appears to be P278S, which is not one of the most common in cancers. In addition, supplementary table 4 shows that two CNAs were identified in donor SG4 (and none in the UK cohort). With regard to TP53 mutation, missense mutations can be more deleterious than copy-number loss due to gain-of-function/dominant negative effect. Combining the observations above with their findings that aggressive mutations, such as those in codon R248, were not observed in Singapore donors, might suggest that more aggressive mutations are negatively selected in the skin of people from Singapore while clones with damaging but less aggressive mutations are not counteracted as efficiently and expand

at levels similar to those seen in people for the UK. Is there evidence that this might be the case? Admittedly, getting into this in the manuscript may be overly speculative.

6. The authors state that “four of the five Singaporean donors and zero UK donors have introgression of the HYAL2 region at chromosome 3p21.” They should briefly refer to the significance of this (a Neanderthal introgression that is associated with UV responses).

7. Is there no sequencing of SCCs from Singaporean subjects available? Such a comparison would have been more ideal.

8. One minor point, on page 3 the authors say “OCA2, which encodes a precursor to the UV-absorbing pigment melanin”. From our understanding OCA2 encodes a protein that is necessary for the synthesis of melanin but is not a precursor of melanin.

Signed: James DeGregori and Marco De Dominicis

Author Rebuttal to Initial comments

Response to Reviewers: Somatic Mutations in Facial Skin from Countries of Contrasting Skin Cancer Risk

We would like to thank referees for the time taken to review our manuscript and are most grateful for their constructive comments which have significantly improved the revised text. Our responses to each point are in blue.

Referee expertise:

Referee #1: skin cancer risk, genetics

Referee #2: skin cancer and ageing

Referee #3: genomics, cell competition (signed report)

Reviewer #1:

The authors have systematically compared the mutational landscape of a large number of biopsies (191) taken from a sun-exposed site from 5 individuals from Singapore. They have contrasted these results with a similar dataset from 6 people from the UK.

The samples from individuals from Singapore showed much lower rates of mutations and smaller/fewer mutant clones.

The overall mutational burden was also very low compared to the samples from the UK, and proportionally lower was the signal from UV-induced mutations even though Singapore experience 2-3 fold higher rates of exposure.

WES of cSCC lesions from 19 Korean individuals (chosen as were similar to the Singapore population) was very similar to the mutation profiles of SCCs from elsewhere in the world suggesting that while the

Figure 4: 36 SNP loci where donor genotype differs by country. SNPs are either associated with keratinocyte cancer (KC), tan response (Tan) or previously found to be under selection in Singaporean genomes (SG); - for genotype indicates a call could not be made (**Methods**). Associated genes and mechanisms are suggested for each SNP using PheWAS and eQTL data from the Open Targets Genetics database. ECM, extra-cellular matrix, - indicates mechanism is unknown. Population allele frequencies are reported for East Asia and Great Britain using data from 1000 Genomes Project Phase 3 (**Methods**). Note all SNPs in the HYAL region form part of the same LD block on chr 3p.21.

We find at least 11 SNPs associated with pigmentation, particularly near to *HERC2*, *MC1R* and *OCA2*, that differ by donor country. *OCA2* encodes a melanosomal anion channel that is essential for melanin synthesis¹ and is under positive selection in Singaporean genomes². A regulatory element within an intron of *HERC2* is known to inhibit the *OCA2* promoter and both these genes have associations with hair, eye and skin pigmentation³. SNPs near *IRF4* show associations with skin tanning, pigmentation and keratinocyte cancer risk (<https://genetics.opentargets.org/gene/ENSG00000137265>). rs12203592-T, which has an allele frequency of 0.18 in the UK but is absent in East Asians, lowers IRF4 levels and hence reduces expression of the pigmentation enzyme encoded by *TYR*⁴. As the reviewer points out, these genetic differences are reflected in the pigmentation and tanning phenotypes of UK and Singaporean skin, assessed by the Fitzpatrick score.

However, differences in pigmentation may not be the only explanation for variation in KC risk between the two countries. In the SNPs that differ by country, at least ten seem to affect KC risk through mechanisms unrelated to pigmentation (**Figure 4**). For example, rs2111485 is associated with inflammatory and autoimmune diseases such as psoriasis, vitiligo, asthma and inflammatory bowel disease, whilst rs12129500 and rs2243289 affect the expression of *IL6R* and *IL4*, respectively.

Other SNPs that differ by country include: rs663743, shown to alter the expression of *PPP1R14B*, a gene highly expressed in multiple tumour tissues with functions related to cell growth, cell cycle regulation and apoptosis (PMID: [34858479](https://pubmed.ncbi.nlm.nih.gov/34858479/)); rs7335046, shown to alter *UBAC2* expression, a gene involved in regulating the Wnt signalling pathway; and rs4409785, which alters *SESN3* expression, a gene involved in cellular response to stress, e.g. reactive oxygen species. Space precludes a discussion of these genes and SNPs in the manuscript.

Finally, we find four of our Singaporean donors have introgression of a large LD block at chromosome 3p21.31. This region is under positive natural selection in East Asians and contains multiple genes associated with tumour suppression (*HYAL1*, *RASSF1*, *TUSC2*), cell motility (*NAA80*) and cellular response to UVB (*HYAL2*) (Ding *et al.*, 2014). We do not propose that the differences observed in the somatic mutational landscape here are due to genes in this region as it is not present across all Singaporean donors in this study, however, we highlight this as an example of a mechanism unrelated to pigmentation which may act to suppress keratinocyte carcinogenesis.

In the revised text we now discuss these genetic differences at two points.

1. In the introduction, emphasising pigmentation differences, as suggested in comment 4 below – **p.2**:

“The incidence of many cancers varies substantially worldwide, reflecting genetic differences between populations and their environmental exposures. This is well-illustrated by keratinocyte skin cancers (KC), where incidence varies 140-fold globally⁵. KC risk increases with an individual’s cumulative UV exposure which depends on age, outdoor work, sunbathing, use of tanning beds⁶⁻¹¹ and phenotypes such as freckles, low levels of skin pigmentation and poor tan response¹². The most strongly associated KC risk loci are found near to pigmentation genes such

as *HERC2*, *OCA2*, *MC1R* and *IRF4*, which have much greater allele frequencies in high-risk populations such as in the UK than in low-risk populations in East Asia¹³.”

2. In discussion of **Figure 4 – p.9-10**:

“To gain insight into differences in genetic background that may exist between donors we genotyped individuals for SNPs associated with KC risk (**Methods, Fig. 4, Table S8**). At 36 risk loci we found differences in donor genotype by country. Multiple SNPs were associated with genes linked to pigmentation, such as *SLC45A2*, *IRF4*, *BNC2*, *OCA2* and *HERC2* (<https://genetics.opentargets.org>). For example, rs12203592 alters *IRF4* levels and expression of the pigmentation enzyme encoded by *TYR*⁴. rs4778210, positively selected in Singaporeans², is adjacent to *OCA2*, encoding a melanosomal anion channel that is essential for melanin synthesis¹. These findings are consistent with the marked difference in UV mutational burden between countries. However, we also observed differences in non-pigmentation or tanning related SNPs. KC risk is strongly linked to immunosuppression and is also associated with inflammatory diseases¹². rs2111485 is associated with inflammatory and autoimmune diseases such as psoriasis and inflammatory bowel disease, whilst rs12129500 and rs2243289 affect the expression of *IL6R* and *IL4*, respectively (<https://genetics.opentargets.org>). We found four of our Singaporean donors have a large introgression at chromosome 3p21.31². This region is under positive natural selection in East Asians and contains the skin tumour suppressor gene *RASSF1*¹⁴ and the UV induced genes *HYALI* and *HYAL2*¹⁵. Other KC risk SNPs were linked to genes with diverse functions.”

Comments

1. Abstract line one, and final sentence page 2. References 1-3, used to report the relative incidences of SCC across Singapore and the UK are from 2005 and earlier - are there no more recent surveys of skin cancer incidence in these countries? Figure 1a shows incidence rates up until 2015 suggesting different sources have been used then were cited (ref 1) in figure 1's footnote;

Suggested action: check/update references

Thank you for highlighting this point. We have now replaced the out-of-date references with those used to generate **Figure 1a**. We have also added a key reference underpinning the UK data¹⁶.

2. The final sentence of the abstract offers no conclusion as to why skin from a low-risk country does not

show multiple features convergent with cancer; the likely answer from the data is the degree of skin pigmentation and propensity to tan. This should be addressed. Also, see points 4 and 11.

Suggested action: Update abstract/text to address

Thank you for this suggestion. We have revised the abstract as below to reflect the likely significance of germline variation in tanning and pigmentation genes as suggested. However, without data on behaviour, we cannot make a conclusive statement on this point. The final two sentences of the abstract now read:

“Ageing skin in a high incidence country has multiple features convergent with cancer, not found in a low-risk country. These differences may reflect germline variation in UV protective genes.”

3. Introduction, paragraph 1, page 2, sentence 5: Refs 18-21 aren't citations to a large twin study (I assume the authors mean Mucci 2016 PMID: 26746459) nor are they all citations to GWAS of KC cancer. 18 is a PRS study, 20 isn't a GWAS, and there are other published KC GWAS other than refs 19 and 21 that should be cited.

Suggested action: check/update references

Thank you. We have rewritten this section of the text, see above, and amended the references.

4. Final sentence of introduction paragraph 1: In my opinion saying that some GWAS SNPs for KC are linked to skin pigmentation somewhat undersells this fact. For example, in ref 21 the majority of the most strongly associated loci (and thus represent the largest combination of effect sizes and minor allele frequency, and thus the proportion of trait variance explained) are all pigmentation SNPs. While other pathways are important in KC risk, pigmentation is by and far the strongest genetic risk for KC, which is why the rates are so elevated in the UK and other countries with majority fair-skinned populations.

Related - in the introduction paragraph 3 while there may be additional genes under selection in East Asian populations that are relevant to skin cancer risk, a likely major reason for the reduced (genetic) risk for KC is the absence or greatly reduced frequency of low pigmentation risk alleles enriched in European ancestry populations e.g.

- rs12203592, a pigmentation SNP influencing IRF4 function and a KC risk SNP has a low pigmentation T allele freq of ~14% in EUR pops, and < 0.01% in East Asian populations.

- rs1805007, one of the strongest red hair alleles and a strong risk allele for KC, has a risk T allele freq of ~7% in EUR but only 0.07% in East Asian populations (source: GNOMAD).

Suggested action: The importance of pigmentation SNPs/genes should be more fully and clearly discussed since it has bearing on their later results.

We agree that the most strongly associated KC risk SNPs are linked to pigmentation and/or tanning and that individuals at very high risk (for example, with alleles such as rs1805007-T) are far more common in Europe. We note that both SG and UK donors in this study lack the rarer but strongly associated KC risk SNPs (rs1805007 and other pigmentation alleles in *MC1R* for example, **Supplementary Table 8**). The UK donors in this study are thus not very high-risk individuals, but nevertheless have multiple risk alleles in common with a Northern European population.

We have updated the text to emphasise the importance of pigmentation on KC incidence. The first paragraph of the introduction has been rewritten, see above. We also highlight the importance of tanning/pigmentation alleles in Figure 4 and the associated discussion, see above.

5. Introduction page 3 - sentence 1. Expand briefly on the behaviour differences with respect to UV exposure and if relevant discuss how it may impact the differences in mutational burden observed.

Suggested action: Update abstract/text to address

Thank you, we have added detail on risk behaviours and now cite a Singaporean study on sun exposure. We have amended the text as follows:

“KC risk increases with an individual’s cumulative UV exposure which depends on age, outdoor work, sunbathing, use of tanning beds⁶⁻¹¹ and phenotypes such as freckles, low levels of skin pigmentation and poor tan response¹².”

6. Introduction page 3 - sentence 2. OCA2 isn’t a precursor to melanin; it is a channel transporter protein most likely involved in maintaining the pH of melanosomes, allowing Tyrosinase’s production of melanin - see PMID 32333855

Suggested action: Update abstract/text to address

Thank-you. We have amended our description to:

“rs4778210, positively selected in Singaporeans², is adjacent to *OCA2*, encoding a melanosomal anion channel that is essential for melanin synthesis¹.”

7. Table 1

- a. It would be very helpful to merge the key mutational load columns from Sup table 12A into this table
- b. Why are the age listed as ranges e.g SG1 is 68-71?
- c. While the mean ages are roughly similar between the UK and SG samples, SG2 is young (28-31) which is relevant given the premise here is to report on ageing skin. The similarity of this sample to the others in terms of mutation load etc should be discussed - e.g. are they an outlier?
- d. The most striking difference in this table is that all the SG samples have higher Fitzpatrick skin types. This makes it difficult to interpret what differences are due to country (and UV exposure) vs host factors (a more photoprotective skin type) as they are confounded. The authors should discuss this and report on the regression of Fitzpatrick score onto mutational load - how much of the variation in mutational load is explicable by Fitzpatrick score?

Suggested action: update tables to address (a), and clarify table or footnote to address (b). (c) and (d) please update analysis/manuscript as required

- a. Thank-you, genome-wide burden estimates have been added to **Table 1**.
- b. Donor ages are listed as ranges to help maintain donor anonymity.

- c. SG2 is ~20 years younger than the youngest UK donor, however, we have not excluded SG2 from the study because donor age is comparable by country (mean: SG = 62 years, UK = 68 years; median: SG = 70 years, UK = 70 years) and there is no evidence to suggest SG2 is an outlier. Although SG2 has the lowest mutation burden of all donors, it is comparable to that of SG3 (72-75 years) and not an outlier (**Fig. 2a, Fig. S1a**).
- d. The Fitzpatrick score is an easily obtained subjective measure of skin UV response and has been very widely adopted, but its application to the global range of skin phenotypes is subject to debate¹⁷. While a regression of Fitzpatrick score by mutational burden is significant (Kruskal-Wallis test: $p = 0.02$), after post hoc testing with multiple testing correction, we find the main contribution to donor mutation burden is country. Furthermore, since Fitzpatrick score correlates with country, it cannot be considered an independent factor. We note that the effect of Fitzpatrick score on burden within Singapore is also non-significant ($p = 0.4$).

Response Figure 1: Fitzpatrick score and mutational burden

We agree that differences in UV responsiveness are likely to be of major importance in shaping the mutational landscape but would submit that including the additional genotypic information in **Figure 4** is more informative than Fitzpatrick score, which we continue to provide in **Table 1** for interested readers.

8. Methods - multiple testing. There are a large number of tests performed - multiple testing correction should be applied and the impact in terms of the significance of results reported.

Suggested action: List and report total number of tests and adjust the manuscript appropriately.

As requested, we now list all tests with their reported significance. Duplicate mentions of tests in both text and figures have been removed from the main text, tests are now only reported in the figure legends with p-values and test names. The tests performed are listed below:

Fig. 2a. Estimates of genome-wide mutation burden per donor by country (Student's t-test: $p = 8.8 \times 10^{-3}$). Here we performed a single t-test and therefore do not require multiple testing correction.

Fig. 2c. We used unsupervised clustering to test if UV signature distributions were different between countries ($p < 1 \times 10^{-5}$). We do not require multiple testing correction.

Supplementary Figure 1: **a** Estimates of genome-wide mutation burden per donor. **b** Proportion of SBS1 mutations/mm² per donor by country (t-test: $p = 0.01$). **c** Proportion of SBS5 mutations/mm² per donor by country (t-test: $p = 2.7 \times 10^{-4}$). **d** Proportion of SBS assigned to each signature, split by mutations above and below median variant allele frequency (VAF) for each country (Pearson's chi-square UK: $p < 2.2 \times 10^{-16}$; SG: $p = 1.3 \times 10^{-14}$). **e** Counts of double-base substitutions (DBS) per mm² of skin of donors from each country (UK mean = 1.08 DBS/mm², SG mean = 0.126 DBS/mm², Welch's t-test: $p = 2.3 \times 10^{-3}$). **f** Count of insertions and deletions per mm² in each donor by country (UK mean = 0.72 indels/mm², SG mean = 0.37 indels/mm², Welch t-test: $p = 0.07$).

Fig S1b. Proportion of SBS1 mutations/mm² per donor by country, t-test: $p = 0.01$. Here we performed a single t-test and therefore do not require multiple testing correction.

Fig S1c. Proportion of SBS5 mutations/mm² per donor by country, t-test: $p = 2.7 \times 10^{-4}$. Here we performed a single t-test and therefore do not require multiple testing correction.

Fig S1d. Proportion of SBS assigned to each signature, split by mutations above and below median variant allele frequency (VAF) for each country, Pearson's chi-square UK: $p < 2.2 \times 10^{-16}$; SG: $p = 1.3 \times 10^{-14}$. No multiple test correction required for the pairwise comparisons within each country.

Fig S1e. Counts of double-base substitutions (DBS) per mm² of skin in donors from each country (UK mean = 1.1 DBS/mm², SG mean = 0.13 DBS/mm², Welch's t-test: $p = 2 \times 10^{-3}$). We use Welch's t-test (for unequal variance), therefore do not require multiple testing correction.

FigS1f. Count of insertions and deletions per mm² in each donor by country (UK mean = 0.72 indels/mm², SG mean = 0.37 indels/mm², Welch t-test: $p = 0.07$). We use Welch's t-test (for unequal variance), therefore do not require multiple testing correction.

Supplementary Figure 2: **a** Ratio of observed/expected non-synonymous mutations for positively selected genes by country ($q < 0.01$). No synonymous mutations were detected in Singaporean skin for *TP53* and *AJUBA*, leading to high dN/dS ratios. Line drawn at $y = 1$. **b** Sizes of all clones with protein-altering mutations in the top four positively selected genes, by country (samples with known CNA removed), t-test adjusted by Bonferroni multiple test correction: $p = 6 \times 10^{-4}$ (*NOTCH1*), 1.5×10^{-2} (*NOTCH2*), 0.68 (*FAT1*) and 1.0 (*TP53*). **c** Distributions of mutations across codons in *TP53*. The most frequently mutated codon in cancer, R248 (shown red), is the most common codon change in UK skin but is absent in Singaporean skin.

Fig. S2b Distribution of clone size for those carrying protein-altering mutations in the top four positively selected genes, by country (samples with known CNA removed). We have added a Bonferroni multiple test correction and reported adjusted p-values, including non-significant values.

Figure 3: Clonal selection and competition differs by country. **a** The number of mutations of each consequence for positively selected genes (dN/dS : $q < 0.01$) by country. *ARID2* is not significantly positively selected in SG samples. **b** Plot of non-synonymous mutations per gene in SG vs. UK samples. Gradient of line = total number of non-synonymous mutations in UK/SG = 6846/1432. Positively selected genes (purple) are labelled. Red indicates positively selected genes with a significant ($p < 0.001$) difference in dN/dS ratio by country, after accounting for global differences. **c** A representation of protein-altering mutations in 1 cm² of skin from donors of Singapore and the UK. Samples were randomly selected and mutations displayed as circles, randomly distributed in the space. Sequencing data, including copy number, was used to infer the size and number of clones and, where possible, the nesting of sub-clones. Otherwise, sub-clones are nested randomly. **d** Estimated percentage of cells with at least one non-synonymous mutation per positively selected gene, by country (samples with known CNA removed). Wilcoxon: $p = 0.03$ (*NOTCH1*), 0.7 (*NOTCH2*), 9×10^{-3} (*FAT1*) and 0.05 (*TP53*), p -adjusted by Holm multiple comparison. **e** Estimated percentage of cells with at least one non-synonymous mutation across 74 genes by country (**Methods**, Wilcoxon: $p = 4.3 \times 10^{-3}$). **f** Violin plot comparing clone size distributions (summed VAF) per mutation by country. Mutations from UK donors had a lower mean summed VAF (Welch's t-test: $p = 4.27 \times 10^{-14}$).

Fig. 3d. Percentage of cells with at least one non-synonymous mutation per positively selected gene, by country, by gene (samples with known CNA removed). We now report the multiple testing correction in the figure legend.

Fig 3e. Percentage of cells with at least one non-synonymous mutation across 74 genes per country. Wilcoxon test, two groups, this does not require multiple testing correction.

Fig 3f. Clone size distributions (summed VAF) per mutation by country. Wilcoxon test, two groups, does not require multiple testing correction.

p.9 Difference in 'hotspot' *TP53* codon distribution (bootstrap test) by country. We tested how likely it was to find a non-zero number of hotspot *TP53* mutations in Singaporean skin, $p < 0.001$.

Methods, p.16.

'There was no evidence to suggest a difference in mean mutation burden between the eyebrows and eyelids of donors (Welch's t-test: $p = 0.14$).'

Welch's t-test, two groups (mutation burden in eyelids and brows per donor), does not require multiple testing correction.

9. Page 8 final paragraph; the justification for removing the large *TP53* clone from SG1 is not clearly presented. Why is it valid to exclude this sample? This should be made clear in the methods/results

Suggested action: Discuss/address in manuscript

The large *TP53* mutant clone (that carries a *NOTCH2* mutation within it) spans 16 2-mm² samples of skin in SG1. It is an outlier in terms of size compared to all other mutations (Fig. S2b, Fig. 3, Table S1). The summed VAF of this clone is 2.86, nearly four times larger than the next largest clone, a *NOTCH1* mutant in a UK donor (summed VAF = 0.75). We note the true size of this *TP53* clone is likely to be larger since it is found on the edge of the piece of tissue sampled (below).

SG1: VAF of *TP53* P278S mutation

Response Figure 2: Spatial map of large *TP53* mutant clone spanning sixteen samples in donor SG1. The VAF for each sample of a single *TP53* P278S mutation is shown. Each rectangle represents a 2x1 mm sample.

We report the percentages of *TP53* mutant tissue by country and the statistical significance both with and without this clone, so the reader is able to interpret both. We have updated the **Methods**.

10. Page 10, second last sentence. “-we would still expect to observe at least 3 mutations at codons R248/R282 across all Singaporean samples” isn’t clear. If I understand it correctly the authors are suggesting 0 observations out of 191 samples is unusual given they expected 3 out of 191 - is that especially unlikely given the events are rare? If that is the correct interpretation is this difference significant using an appropriate non-parametric test? If not clarify this result.

Suggested action: Clarify in manuscript and include additional results if relevant

Thank-you for this helpful suggestion. We applied non-parametric bootstrapping to estimate the significance of observing 0 mutations at R248 and R282 *TP53* codons in Singaporean skin.

Response Figure 3: Bootstrap test to estimate the significance of observing 0 mutations at R248 and R282 *TP53* codons in Singaporean skin. Of 1,000 simulations, none produced either an observed value

of zero (green = bootstrapped estimation from each sub-sampling, red = Singaporean observation) or the 95% upper confidence interval limit (0.0156).

We describe this test in the **Methods** as follows:

“We applied non-parametric bootstrapping to estimate the significance of observing 0 mutations at R248 and R282 *TP53* codons in Singaporean skin, given the decreased burden, decreased *TP53* selection and decreased UV signature compared to UK skin. The non-parametric bootstrap statistical method is appropriate for this data because it does not make assumptions on distribution¹⁸. The method samples from a given distribution (here, UK skin), maintaining variability and other observable parameters. For a bootstrap test we simulated expected values for mutation distribution in Singaporean skin given: (i) UK mutation distribution per donor and per sample of all, C>T/CC>TT, *TP53* and R248W/R282W *TP53* mutations and (ii) proportions of all, C>T/CC>TT and *TP53* mutations between UK and SG. If the null hypothesis that the similarity of UK (adjusted) and SG distributions is true, we would expect that the simulated ‘UK-adjusted’ values would correspond to the SG observations, on average.

We applied the ‘rule of three’ approach, often used in clinical trials,¹⁹ to estimate the robustness of zero outcomes in Singaporean skin. This approach suggests the upper limit of a 95% confidence interval is $3/(n+1)$, where n is the number of SG samples. Here, this value is $3/(191+1)= 0.0156$. We ran 1000 simulations to obtain estimation for an event of zero R248/R282 *TP53* codon mutation counts in SG samples. Of 1,000 simulations, none produced a value lower than 0.0156. The bootstrap simulation shows that (i) an expected mutation frequency at R248/R282 *TP53* codons is around 3 mutations across all SG samples and (ii) the UK and SG distributions are significantly different ($p < 0.001$).”

We now state in the manuscript **p.9**:

“Even after adjusting for C>T/CC>TT burden across *TP53* by country, we would still expect to observe approximately 3 mutations at codons R248/R282 across all Singaporean samples (Methods, Bootstrap test, $p < 0.001$)”.

11. Figure 4 indicates that the SG population essentially lacks any of the major pigmentation risk alleles. This is interesting (but perhaps not surprising). In general the implications of this aren't really addressed in the manuscript, and relates to point 4 - that the likely main reason the SG samples have lower mutation burdens than UK samples is they have greater skin pigmentation and that greater protection against UV damage. Specific comments about the SNP selection and figure 4:

- a) Please include chr and position and present SNPs in their chr/bp order.
- b) Please include an appropriate reference population allele freqs for EUR and EA (e.g. from GNOMAD) as a column to help contextualise the results. This is relevant because many SNPs are rare (e.g. rs192481803 has a MAF ~ 0 in all pops which is likely why everyone is missing the risk allele) and/or many of the risk alleles related to pigmentation are essentially absent in East and South Asian populations e.g. see point 4.
- c) Please check if the listed SNPs are in LD with each other at least in EUR pops (as their BCC/SCC associations were identified in EUR ancestry pops) e.g. rs4268748 is an intronic variant near MC1R, and is in LD with the rs1805007 functional missense allele (LD r^2 0.23 in EUR) which is likely why it has previously been reported for BCC/SCC. As rs1805007 is wildtype in SG samples I am not sure it is relevant this SNP is heterozygous in some SG samples. The authors should consider removing SNPs in LD with other functional SNPs if they exist.
- d) Some SNP IDs are out of date e.g. the ID rs74664507 has been merged with rs11445081
- e) Some listed SNPs have known functions/associations assigned since the publication of the cited reviews (41 and 42) e.g. rs74664507 / rs11445081 is strongly associated with skin colour. The authors should survey more recent literature, or databases like Open targets genetics (e.g. https://genetics.opentargets.org/variant/9_16913838_T_A [genetics.opentargets.org]) to update the figure to better reflect current information.

Suggested action: update/check SNP data entered into figure 4 and related to points 2 and 4 address implications in the manuscript.

Thank-you, all revised – please see **Figure 4** and the SNP analysis above. We present SNPs in chr/bp order with East Asian and GBR allele frequencies (1000 Genomes Project Phase 3). We note that allele frequencies for SouthEast Asia or Singapore are unavailable. East Asia allele frequencies are calculated from 504 individuals (~ 300 Chinese, ~ 100 Japanese and ~ 100 Vietnamese). GBR allele frequencies are taken from 91 individuals across England and Scotland. We have kept all SNPs that we genotyped across the HYAL region in **Figure 4** but have highlighted in the legend that they form the same LD block. As the reviewer recommended, we have updated associated genes and phenotypes using the Open Targets Genetics database.

12. Methods - page 17. Please provide more details on the reference panel used to call mutations. Specifically, what was the ancestry of the reference population used, and did it match each sequenced population e.g. using an off-ancestry reference panel could be expected to increase the mutation detection rate due to real SNP differences between panel and target.

Suggested action: update methods and discuss if relevant.

The reference panel for deepSNV is used to construct a base by base model of the background error rate of the sequencing platform²⁰⁻²². This allows the identification of loci that have a statistical excess of mismatched base calls, enabling the reliable detection of low VAF mutations. To generate the large reference panel, 51 deeply sequenced samples of muscle or fat were used all from donors of the UK. Variants are called if the VAF exceeds the error rate at the base concerned.

The reviewer raises the issue of germline variants. These would be present in all samples (n = 34-45) in a given donor at a high VAF, allowing us to exclude any such variants as germline. We have now explained these points in more detail in the **Methods**.

13. Methods page 18 paragraph 1 final sentence + sup figure 3. Please report the significance value for the correlation $R = 0.33$ between coverage and mutations detected.

Suggested action: Report significance, and if this is significant the statement starting “no evidence” will need to be adjusted and this limitation addressed in the discussion

Thank you. The significance of the correlation for the combined SG and UK data shown in the original **Fig. S3** is high (p is very small). We note, a statistically significant correlation coefficient does not mean there is a strong association. It tests the null hypothesis that there is no relationship but provides no information about the strength of the relationship or its importance. The relationship between two variables is generally considered strong when r is greater than 0.7²³.

We have, however, incorrectly calculated the correlation in the original **Fig. S3** due to two statistical pitfalls in combining two different populations (UK and Singapore)²⁴, these are:

- (i) when data has two subgroups, within each of which there is no correlation
- (ii) when variability in values on the x-axis changes with values on the y-axis (and we note that variance grows with coverage here)

We do indeed have both (i) and (ii) in our original approach. Plotting within each group (see amended **Fig. S3** below), we show there is no significant correlation:

Supplementary Figure 3: Correlation between mean quality sequencing coverage per sample and the number of mutations detected, by country. Mean coverage was calculated after removing off-target reads, duplicates and those with mapping quality of 25 or less and base quality of 30 or less. Samples of neither country show a correlation between coverage and mutation counts (SG: $R = 0.087$, $p = 0.089$; UK: $R = 0.092$, $p = 0.058$).

We are most grateful to the reviewer for leading us to correct this error.

The aim of **Fig. S3** is to show that even at the ranges where the coverage is equal (700 to 1000x), more mutations are called in UK samples. The Shearwater algorithm (deepSNV package) does not require checking of any correlation beyond ensuring a minimum coverage of more than 500x. Above this level, there is enough statistical power for an unbiased variant call, so our correlation is perhaps unnecessary.

However, for completeness, we have replaced the misleading original **Fig. S3** with separate panels for each data set. This demonstrates that when samples are grouped appropriately, there is no correlation between coverage and mutation calls and, most importantly, that coverage exceeds 500x in both groups.

We also now state in the **Methods p.15**:

“We filtered sequencing reads for quality by mapping and base quality. Mean quality sequencing depth of coverage exceeded 500x per sample in both Singapore and UK samples and the number of mutations detected did not correlate with sequencing depth in either group (**Fig. S3**).”

14. Methods video model + model images. Are these simulations derived in any way from the data other than starting the mutation burden 4 fold higher in video 2? E.g. were the model's settings for fitness derived from the reported dataset? If so please state so clearly. If not, please clarify what these videos are showing us about the data presented, or how they are otherwise providing nuance to the manuscript.

Suggested action: Clarify methods for these videos and their relevance to the paper

The videos aim to illustrate the effect of mutation burden on clone size distributions. We show that, with a 4-fold higher mutation burden, clones are more restricted in their growth, leading to a smaller clone size distribution (as observed in the UK skin samples). It may be counter-intuitive for some readers that clone sizes are larger in Singaporean skin compared to the UK, so the aim of the video is to aid understanding by illustrating our hypothesis: high competition in UK skin leads to restriction of clone growth.

We have amended the **Methods** to now read:

“We illustrated mutant clone growth in a sparsely (**Video 1**) and densely (**Video 2**) mutated environment to simulate clonal competition in skin from Singapore and the UK, respectively. In Video 1, mutant clones of two arbitrary levels of fitness divide against a wild-type background of lower fitness until approximately 50% of the space is mutant. The starting mutation burden of Video 2 is 4-fold higher than for Video 1, with cells dividing for the same number of divisions. Final mutant clone sizes are larger in a sparsely mutated environment (**Video 1**) compared to in a densely mutated environment (**Video 2**), reflecting the different clone size distributions we observe by country (**Fig. 3f**).”

Reviewer #2:

This paper compares the accumulated mutations in human epidermis of individuals of two different ethnic backgrounds, UK and Singapore. The two ethnicities have a different lifetime risk of developing cSCC, independently of UV exposure. The comparison shows individuals from the high-risk ethnicity (UK) have a distinct pattern of mutations and mutational processes that overlap with cancer-driving cSCC genes, and individuals from the cSCC low risk group (Sing) in contrast have fewer mutations, fewer mutations in known SCC driver genes, and importantly, have an increase in the mutations that are linked to skin ageing but not skin cancer.

This study is important, novel and opens many new questions that will impact scientists and clinicians working in the life sciences, not only cancer biology. The potential impact to medicine is clear.

The main limitation is there are no valid hypotheses put forward to explain the difference in mutational landscape between UV exposed skin of different ethnic backgrounds, no experimental work to explain why different ethnic backgrounds develop/select different mutations.

Questions

a. Previous work shows sbs1 sig reflects the number of cell divisions, as it increases at a constant rate per cell division. The rate of cell division in human skin depends on sun exposure and age. Can the

authors infer whether the rate of sbs1 mutations over time is equal in both skin types, taking into account sbs7? It is interesting to test whether cell cycle regulation underpins ethnic differences.

SBS1 has indeed been argued to reflect the number of cell divisions in earlier work, though the progressive increase of SBS1 mutations with age in post-mitotic cells such as neurons and smooth muscle cells challenges this assumption^{25,26}.

The number of SBS1 mutations/mm² of skin is significantly higher in the UK than in Singapore (**Response Figure 4**). Dividing by donor age gives mean rates of SBS1 mutation of 0.0006/year/mm² and 0.010/year/mm² in Singapore and the UK respectively.

Response Figure 4: Number of SBS1 per mm² by country (t-test: $p = 0.01$).

One way to explore the impact of cell division on signatures is to consider clone size. We examined whether signatures were different in clones smaller or larger than the median VAF, finding this was the case in both countries (Pearson's chi-square UK: $p < 2.2 \times 10^{-16}$; SG: $p = 1.3 \times 10^{-14}$). The proportion of SBS1 mutations is higher in larger clones than in smaller clones in both countries.

Response Figure 5: Proportion of SBS assigned to each signature, split by mutations above and below median VAF for each country.

In terms of adjusting for SBS7 we find that, across both countries, SBS1 (and SBS5) are positively correlated with SBS7 (Pearson's $R = 0.90$ and 0.91 , respectively). This might reflect clonal expansion driven by UV light.

We thank the reviewer for suggesting this line of enquiry. The increased proportion of SBS1 in larger clones suggests SBS1 may increase with increased cell division in larger clones, though these may also be older in chronological time. The uncertainty over the link between cell division and SBS1 perhaps argues for a cautious interpretation of this data. We present this result in the text **p.6** as follows:

“The number of both SBS1 and SBS5 mutations/mm² was increased in UK skin compared with Singaporean skin (**Fig. S1b-c**). In epithelial cells, the proportion of SBS1 mutation has previously been correlated with the cumulative number of cell divisions in a tissue. Here, in both countries, we find increased SBS1 in larger clones (**Fig. S1d**). These differences may reflect the effects of UV, which increases the rate of proliferation and proportion of dividing cells in the epidermis and can drive mutant clone expansion²⁷”

b. If there are differences in the cell division rate inferred from the sbs1 rate, adjusted by age and sbs7, for example faster cycling in UK skin, could these differences link to less time for DNA repair, i.e more mutations? Have they compared DNA repair?

The burden of SBS1 and SBS5 mutations correlate with SBS7 burden (Pearson's $R = 0.90$ and 0.91 , respectively). As discussed above, UV driven clonal expansion may contribute to the increased number of SBS1 mutations in UK skin.

We are limited in our ability to compare DNA repair beyond analysis of mutational signatures. It is hypothesised that SBS7a and SBS7b are each the consequence of the two major known UV photoproducts: cyclobutane pyrimidine dimers and 6-4 photoproducts, whilst SBS7c and SBS7d may be the consequence of translesion DNA synthesis by error-prone polymerases inserting T or G respectively, rather than A, opposite UV induced photodimers. SBS7d, is increased in UK skin as expected with increased UV exposure.

C>T mutations caused by UV damage are more frequently observed on the untranscribed strand of genes, due to the repair of lesions on the transcribed strand by transcription-coupled nucleotide excision repair²⁸. However, C>T mutations caused by SBS1 do not exhibit strong transcriptional strand bias²⁹. There is a stronger transcriptional strand bias of C>T mutations in skin from donors of the UK compared to Singapore, consistent with increased UV damage in the UK (transcribed/untranscribed = 0.829 and 0.896, respectively). This is consistent with less repair of UV damage in UK skin, compared to Singapore.

c. You propose that notch1 can prevail and expand in Singapore skin due to less competition from other clones. You propose this is a possible explanation for the difference in clone selection. This possibility could be confirmed in skin of young UK people, with fewer mutations. Does younger UK skin have high notch1, and will ageing reduce these in favour of the more UK-prevalent p53 clones? My hypothesis is that this will not be the case, and young skin will select R348W and R282W TP53. There are no alternative hypotheses in the paper to explain why the accumulation of specific mutations is not stochastic across both skins.

Unfortunately, as blepharoplasty surgery is performed on ageing skin, it is not possible for us to sample facial skin from young UK donors. However, we have attempted to address the reviewer's question through analysis of previously published samples of skin from UK donors that have been taken from less sun-exposed sites²¹.

The Singaporean samples carry 1-28 mutations per sample. We identified 7 previously published UK donors with less than 30 mutations per sample in other body sites.

Response Figure 6: The number of mutations detected through previously published targeted sequencing (74 genes) across skin samples from 7 UK donors and three body sites (Leg, Trunk and Ab = Abdomen).

In total, 1,658 clones are detected across these donors (ages 26-68) in Abdominal, Trunk and Leg skin which we can compare with the 1,839 clones detected across the Singaporean donor facial skin.

Response Figure 7: a Mutational signature decomposition for three UK donors with >200 single-base substitutions using SigProfiler. **b** Ratio of observed/expected non-synonymous mutations for positively selected genes by country ($q < 0.01$), using dNdScv.

Only three of the UK donors had enough mutations (>200) for reliable signature calling. Despite similar mutation burdens to SG donors, the majority of mutations in the additional UK samples were caused by UV damage. This likely reflects germline differences in SNPs related to UV response, discussed below in point d, between the UK and Singaporean populations.

Four genes were identified as positively selected in the low burden UK skin (*NOTCH1*, *NOTCH2*, *FAT1* and *TP53*).

Response Figure 8: a The number of mutations of each consequence for positively selected genes (dNdScv: $q < 0.01$) between SG facial skin and low-burden UK body skin. **b** Plot of non-synonymous mutations per gene in SG facial vs. UK body samples. Positively selected genes (purple) are labelled. Red indicates positively selected genes with a significant ($p < 0.001$) difference in dN/dS ratio by country, after accounting for global differences.

Mutant *NOTCH1* is more strongly selected in Singaporean skin compared to UK skin of similar burden, while mutant *FAT1* is more strongly selected in UK skin (however, we note that mutant *FAT1* was reported as depleted in facial skin in pan-body analysis)²¹.

Response Figure 9: Sizes of all clones with protein-altering mutations in positively selected genes, by country (SG facial skin vs. UK body skin).

Amongst the selected mutant genes, clone sizes in the UK low burden skin were consistently smaller than in the UK. There are multiple potential explanations for this, including body site (Singaporean skin is exposed to UV regularly whereas UK skin is in intermittently sun exposed sites) and donor age (UK donors being younger than Singaporean).

Across the 225 UK low-burden samples, we find two *TP53* R248W mutations (36 year old abdominal skin and 30 year old trunk) and one occurrence of *TP53* R282W (68 year old abdominal skin).

Unfortunately, variation in body site, age and genetic background confound comparison between UK and Singapore and we would submit that discussion of this analysis in the revised text is overly speculative for these reasons.

d. Is there any evidence ethnic loci risk scc variants promote selection of specific DNA mutations? Can this be tested?

e. Many of the high risk loci are linked to pigmentation. Is the rationale that Singapore skin will make “more” pigment to protect better from UV? In reality, Singapore skin has a similar level of pigmentation compared to UK skin, at least the difference is not large enough to explain the dramatic difference in incidence. Moreover, many Mediterranean countries have extremely high incidence of scc, compared to Singapore, and these differences cannot be explained by pigmentation level alone. Do they have any mechanistic insight to what any of the variants do?

Points d and e are both interesting and important issues related to donor genotype. We selected SNPs from the NHGRI-EBI GWAS and Open Targets databases and combined loci associated with keratinocyte carcinoma (entry EFO_0010176 - keratinocyte carcinoma), tan response (EFO_0004279 – suntan) and ‘Ease of skin tanning’ (UK Biobank: 1727). We added these to 16 positively selected loci in Singaporean genomes (Wu *et al.*, 2019, **Supplementary Table 8**).

We have updated **Figure 4** to highlight genotypes which differ most between donors of each country:

Regarding **point d**, we do not have the statistical power to determine if these mutations alter the selection of specific mutant genes.

Addressing **point e**, we have added the following to the text along with the revised Figure 4:

“To gain insight into differences in genetic background that may exist between donors we genotyped individuals for SNPs associated with altered risk for KC (**Methods, Fig. 4, Table S8**). At 36 risk loci we found differences in donor genotype by country. Multiple SNPs were associated with genes linked to pigmentation, such as *SLC45A2*, *IRF4*, *BNC2*, *OCA2* and *HERC2* (<https://genetics.opentargets.org>). For example, rs12203592 alters IRF4 levels and expression of the pigmentation enzyme encoded by *TYR*⁴. rs4778210, positively selected in Singaporeans², is adjacent to *OCA2*, encoding a melanosomal anion channel that is essential for melanin synthesis¹. These findings are consistent with the marked difference in UV mutational burden between countries. However, we also observed differences in non-pigmentation or tanning related SNPs. KC risk is strongly linked to immunosuppression and is also associated with inflammatory diseases¹². rs2111485 is associated with inflammatory and autoimmune diseases such as psoriasis and inflammatory bowel disease, whilst rs12129500 and rs2243289 affect the expression of *IL6R* and *IL4*, respectively (<https://genetics.opentargets.org>). We found four of our Singaporean donors have a large introgression at chromosome 3p21.31². This region is under positive natural selection in East Asians and contains the skin tumour suppressor gene *RASSF1*¹⁴ and the UV induced genes *HYAL1* and *HYAL2*¹⁵. Other KC risk SNPs were linked to genes with diverse functions.”

Reviewer #3:

We know that skin cancer risk varies enormously across populations around the world. But the question is – why? King and colleagues present an analysis of the somatic mutational landscapes of eyelid skin from subjects from Singapore (~17X lower skin cancer risk compared to UK, despite much higher UV exposure), comparing results from a previously reported UK cohort. The use of eyelid samples should control for differences in skin coverage with clothing, with the caveat that sunglasses can block UV exposure. Their studies reveal striking differences in mutational burden and signatures, selection, and mutational hotspots in expansions across the two cohorts, with the UK skin exhibiting greater features found in skin cancers. Results are very clearly presented and well written. Data are supportive of claims, which are not overstated. Methods are well described. One caveat of the study is that they are comparing two cohorts that differ both by genetics and by sun/UV exposure, making it more difficult to ascertain relative contributions to the mutational patterns observed. Still, it is interesting that mutational burden and “pro-cancer” selective patterns are greater for the UK subjects despite substantially less overall UV exposure, perhaps relating to greater genetic risk (resulting in less protection by skin pigmentation).

In all, this study makes an important contribution to the field of somatic mutations in normal tissues and how they could contribute to cancer risks, including risks that vary across populations. Major findings include:

1. Mean mutational burden is 4X higher in UK subjects than Singapore subjects, and the UK subjects had substantially greater UV mutational signatures and 8X as many double base substitutions characteristic of UV exposure. CNAs were evident in 13% of UK eyelid specimen compared to just 1.0% of Singaporean. Thus, mutational burden correlates with skin cancer risk.

2. Perhaps one of the most interesting findings from this study is that fitness landscapes appear to differ for the two populations, with NOTCH1 and NOTCH2 clonal expansions occupying less and TP53 expansions occupying more of the epidermis for the UK subjects than expected by the mutational burden. For the TP53 expansions, this was only true if an outlier clone was removed from one Singaporean subject (a reasonable exercise). In all, while about half of the epidermis in the Singapore group is occupied by clones with disruptive mutations in at least one of the 74 examined genes, this number is about 95% for the UK cohort. The distribution of clone size is smaller for the UK cohort, perhaps indicating clonal competition, which may also explain why NOTCH1/2 clones are significantly smaller (suppressed) in the UK skin samples (in contrast, TP53 clone size was similar across the cohorts). This competition is illustrated in 2 videos of a computational model.

3. They observed differences in the mutational spectra for TP53, FGFR3 and RAS genes, with the mutations most commonly observed in skin cancers (or cancers in general) almost exclusively restricted to the UK cohort.

4. Not surprisingly, genetic (germline) variants associated with pigmentation and SCC risk were differentially found in the UK cohort. Other risk alleles showed similar distributions across the two cohorts.

5. Analyses of cSCC from South Korea (where risk of SCC is similar to that of Singapore) revealed that the UV mutational signature still dominates. If we can extrapolate to SCCs from Singapore (not ideal, but not totally unreasonable), this suggests that despite lower mutational burden in the skin and a reduced UV signature in Southeast Asians, that the cancers are still driven by UV mutagenesis.

In all, their data support their conclusion that UK skin shares more features with skin cancers (in terms of mutational burden and signatures, CNAs, genes under selection, and hotspot mutations) compared to Singaporean skin.

Questions to address:

1. Do Brits tend to holiday in warmer places with more sun exposure, and perhaps experience sun burns? Periodic high exposure may be worse than regular moderate exposure, as the skin can adapt to the latter. Could increased vacationing of Brits closer to the equator have contributed to the 2-3X rise in skin cancer incidence over the last few decades?

The rise in overseas holidays to the Mediterranean region is indeed one of the behavioural factors that is suspected to be responsible for the increasing incidence of keratinocyte cancer, but multiple other behaviours are also likely to contribute, such as the use of indoor tanning facilities and lack of awareness and/or adoption of 'safe sun' practices in previous decades³⁰⁻³². Keratinocyte cancer risk reflects cumulative UV exposure over a lifetime as well as episodes of sunburn^{8,9}. The accurate measurement of UV exposure over such long periods makes research in this area very challenging and so we have

chosen not to speculate on the multiple possible behavioural differences that may lie behind the different mutational landscapes in the UK and Singapore.

2. Were the UK donors of North European descent (Anglo-Saxon) and the Singapore donors of South Asian descent? It's important to make this clear, as there has clearly been a lot of migration of humans.

All Singaporean donors self-reported their ethnicity as either Chinese or Malay. We report, where available, the Fitzpatrick score of each donor in **Table 1** as a phenotypic measure of skin type and taking this, and our analysis of donor genotype, we believe our donors are typical of the ancestries of a small sample of each country's population.

3. From Supplementary table 3 it can be observed that the number of non-UV associated SBS1 and SBS5 mutations are more than twice in the UK cohort compared to the Singapore cohort – this should be pointed out in the manuscript. One interpretation of this observation is that the skin of UK subjects is more prone to mutation-driven clonal expansions independent of UV mediated mutagenesis, such as UV induced selection; alternatively, these mutations could be due to indirect UV mutagenesis or other exposures.

Both SBS1 and SBS5 are associated with tissue ageing and are ubiquitous amongst normal tissues and cancers in humans. SBS1 is caused by the endogenous deamination of 5-methyl-cytosine to thymine and rates of SBS1 acquisition in different cell types correlate with estimated rates of stem cell division²⁹. After adjusting for the area sampled in each donor, there are indeed significantly more SBS1 and SBS5 mutations in UK skin than in Singaporean skin.

Response Figure 10: Proportion of SBS1 and SBS5 mutations/mm² per donor by country.

This is an interesting finding. The reviewer's hypothesis that SBS1 and SBS5 mutations are more likely to be expanded in UK skin, through mutation and/or UV-driven clonal expansion is plausible. In both the UK and Singapore, we find proportions of SBS signatures vary with clone size, see figure below. In both countries, SBS7a was over-represented in mutations below the median VAF and SBS1 was over-

represented in mutations above the median VAF. It is possible that larger clones have undergone more cell division, leading to an increase of SBS1 mutations.

Response Figure 11: Proportion of SBS assigned to each signature, split by mutations above and below median VAF for each country (Pearson’s chi-square UK: $p < 2.2 \times 10^{-16}$; SG: $p = 1.3 \times 10^{-14}$).

We have added the plots above to **Supplementary Figure 1** and update our discussion **p.6** in the manuscript to now read:

“The number of both SBS1 and SBS5 mutations/mm² was increased in UK skin compared with Singaporean skin (**Fig. S1b-c**). In epithelial cells, the proportion of SBS1 mutation has previously been correlated with the cumulative number of cell divisions in a tissue. Here, in both countries, we find increased SBS1 in larger clones (**Fig. S1d**). These differences may reflect the effects of UV, which increases the rate of proliferation and proportion of dividing cells in the epidermis and can drive mutant clone expansion²⁷”

4. For Fig S2a, there seems to be a substantial difference in dN/dS for missense and nonsense+splice TP53 mutations comparing Singapore and UK subjects, both of which are much higher in the Singapore subjects. Could this just reflect the overall lower mutational burden in the Singapore cohort, and thus when a TP53 mutation is detected it is more likely to alter the protein? Some speculation might be warranted, unless we are just missing something.

No synonymous mutations were detected in *TP53* in Singaporean skin, leading to the high dN/dS ratios (**Supplementary Table 5**). This reflects a relative depletion of synonymous mutations across all genes in Singaporean skin compared with the UK (Wilcoxon: $p = 0.0043$). We may speculate that synonymous clones in the skin are likely to follow neutral drift and are far less likely than positively selected mutants to reach the lower limit of detection¹⁸, most synonymous clones that are detected are passengers in mutant clones under selection. If this is the case it would explain why there are more synonymous mutations in

UK skin in which the density of selected mutant clones, and hence synonymous mutant passengers, will be higher. We submit this is perhaps too speculative an argument for inclusion in the text, however.

5. Also, with regard to TP53, while it's reasonable to assume that the large TP53 clone that is observed in donor SG1 is an outlier, it still begs the question if it might have some biological significance. The mutation in question appears to be P278S, which is not one of the most common in cancers. In addition, supplementary table 4 shows that two CNAs were identified in donor SG4 (and none in the UK cohort). With regard to TP53 mutation, missense mutations can be more deleterious than copy-number loss due to gain-of-function/dominant negative effect. Combining the observations above with their findings that aggressive mutations, such as those in codon R248, were not observed in Singapore donors, might suggest that more aggressive mutations are negatively selected in the skin of people from Singapore while clones with damaging but less aggressive mutations are not counteracted as efficiently and expand at levels similar to those seen in people for the UK. Is there evidence that this might be the case? Admittedly, getting into this in the manuscript may be overly speculative.

Thank you for generating such an attractive hypothesis. To explore this, we compared the clone sizes, inferred from the summed variant allele frequency, of missense TP53 mutants in both countries, comparing the size distribution of clones carrying canonical 'hotspot' TP53 mutant codons with other missense mutations. This assumes clone size is an indicator of competitive fitness for mutants within the same gene in the same country.

Unfortunately, there is insufficient evidence to suggest the non-hotspot TP53 mutations generate larger clones than hotspot TP53 mutants in Singapore or in the UK in our data ($p=0.2$ by non-parametric ANOVA with random effect).

Response Figure 12: Clone size distribution for canonical 'hotspot' missense mutations of TP53, and other (nonhot) missense mutations, $p = 0.2$, non-parametric ANOVA with random effect in Singaporean skin.

6. The authors state that "four of the five Singaporean donors and zero UK donors have introgression of the HYAL2 region at chromosome 3p21." They should briefly refer to the significance of this (a Neanderthal introgression that is associated with UV responses).

We have amended our discussion of **Figure 4** to now read (**p.10**):

“We found four of our Singaporean donors have a large introgression at chromosome 3p21.31². This region is under positive natural selection in East Asians and contains the skin tumour suppressor gene *RASSF1*¹⁴ and the UV induced genes *HYAL1* and *HYAL2*¹⁵.”

7. Is there no sequencing of SCCs from Singaporean subjects available? Such a comparison would have been more ideal.

We agree, but unfortunately DNA sequencing of skin cancers in non-European ancestry populations is sparse and, at time of writing, no keratinocyte skin cancer sequencing is available for patients from Singapore.

8. One minor point, on page 3 the authors say “*OCA2*, which encodes a precursor to the UV-absorbing pigment melanin”. From our understanding *OCA2* encodes a protein that is necessary for the synthesis of melanin but is not a precursor of melanin.

Thank-you for correcting us on this point. We have changed our description to:

“rs4778210, positively selected in Singaporeans², is adjacent to *OCA2*, encoding a melanosomal anion channel that is essential for melanin synthesis¹.”

Signed: James DeGregori and Marco De Dominici

Response References

1. Bellono, N.W., Escobar, I.E., Lefkovith, A.J., Marks, M.S. & Oancea, E. An intracellular anion channel critical for pigmentation. *Elife* **3**, e04543 (2014).
2. Wu, D. *et al.* Large-Scale Whole-Genome Sequencing of Three Diverse Asian Populations in Singapore. *Cell* **179**, 736-749.e15 (2019).
3. Visser, M., Kayser, M. & Palstra, R.J. HERC2 rs12913832 modulates human pigmentation by attenuating chromatin-loop formation between a long-range enhancer and the OCA2 promoter. *Genome Res* **22**, 446-55 (2012).
4. Praetorius, C. *et al.* A polymorphism in IRF4 affects human pigmentation through a tyrosinase-dependent MITF/TFAP2A pathway. *Cell* **155**, 1022-33 (2013).
5. Sung, H. *et al.* Global Cancer Statistics 2020: GLOBOCAN Estimates of Incidence and Mortality Worldwide for 36 Cancers in 185 Countries. *CA: A Cancer Journal for Clinicians* **71**, 209-249 (2021).
6. Veierød, M.B., Couto, E., Lund, E., Adami, H.-O. & Weiderpass, E. Host characteristics, sun exposure, indoor tanning and risk of squamous cell carcinoma of the skin. *Int. J. Cancer* **135**, 413-422 (2014).
7. Ferrucci, L.M. *et al.* Indoor tanning and risk of early-onset basal cell carcinoma. *J. Am. Acad. Dermatol.* **67**, 552-562 (2012).
8. Wu, S. *et al.* Cumulative ultraviolet radiation flux in adulthood and risk of incident skin cancers in women. *British Journal of Cancer* **110**, 1855-1861 (2014).
9. Savoye, I. *et al.* Patterns of Ultraviolet Radiation Exposure and Skin Cancer Risk: the E3N-SunExp Study. *Journal of Epidemiology* **28**, 27-33 (2018).
10. Kolitz, E. *et al.* UV Exposure and the Risk of Keratinocyte Carcinoma in Skin of Color: A Systematic Review. *JAMA Dermatol* **158**, 542-546 (2022).
11. Cheong, K.W., Yew, Y.W. & Seow, W.J. Sun exposure and sun safety habits among adults in Singapore: a cross-sectional study. *Ann Acad Med Singapore* **48**, 412-28 (2019).
12. Nagarajan, P. *et al.* Keratinocyte Carcinomas: Current Concepts and Future Research Priorities. *Clinical Cancer Research* **25**, 2379-2391 (2019).
13. Auton, A. *et al.* A global reference for human genetic variation. *Nature* **526**, 68-74 (2015).
14. Tommasi, S. *et al.* Tumor susceptibility of Rassf1a knockout mice. *Cancer Res* **65**, 92-8 (2005).
15. Rauhala, L. *et al.* Low dose ultraviolet B irradiation increases hyaluronan synthesis in epidermal keratinocytes via sequential induction of hyaluronan synthases Has1-3 mediated by p38 and Ca²⁺/calmodulin-dependent protein kinase II (CaMKII) signaling. *J Biol Chem* **288**, 17999-8012 (2013).
16. Venables, Z.C. *et al.* Epidemiology of basal and cutaneous squamous cell carcinoma in the U.K. 2013-15: a cohort study. *Br J Dermatol* **181**, 474-482 (2019).

17. Goon, P., Banfield, C., Bello, O. & Levell, N.J. Skin cancers in skin types IV-VI: Does the Fitzpatrick scale give a false sense of security? *Skin Health Dis* **1**, e40 (2021).
18. Gareth James, D.W.T.H.R.T. *An introduction to statistical learning : with applications in R*, (New York : Springer, [2013] ©2013, 2013).
19. Hanley, J.A. & Lippman-Hand, A. If nothing goes wrong, is everything all right? Interpreting zero numerators. *Jama* **249**, 1743-5 (1983).
20. Gerstung, M., Papaemmanuil, E. & Campbell, P.J. Subclonal variant calling with multiple samples and prior knowledge. *Bioinformatics* **30**, 1198-1204 (2014).
21. Fowler, J.C. *et al.* Selection of Oncogenic Mutant Clones in Normal Human Skin Varies with Body Site. *Cancer Discov.* **11**, 340-361 (2021).
22. Martincorena, I. *et al.* Somatic mutant clones colonize the human esophagus with age. *Science* **362**, 911-917 (2018).
23. Michiels, B. Statistics at square one, 10th edn. *T D V Swinscow, M J Campbell. (Pp 158; £11.95). BMJ Books, 2002. ISBN 0-7279-1552-5. 57, 542-542 (2003).*
24. Aggarwal, R. & Ranganathan, P. Common pitfalls in statistical analysis: The use of correlation techniques. *Perspect Clin Res* **7**, 187-190 (2016).
25. Alexandrov, L.B. *et al.* Clock-like mutational processes in human somatic cells. *Nature Genetics* **47**, 1402-1407 (2015).
26. Abascal, F. *et al.* Somatic mutation landscapes at single-molecule resolution. *Nature* **593**, 405-410 (2021).
27. Murai, K. *et al.* Epidermal Tissue Adapts to Restrain Progenitors Carrying Clonal p53 Mutations. *Cell Stem Cell* **23**, 687-699.e8 (2018).
28. Duan, M., Speer, R.M., Ulibarri, J., Liu, K.J. & Mao, P. Transcription-coupled nucleotide excision repair: New insights revealed by genomic approaches. *DNA Repair* **103**, 103126 (2021).
29. Alexandrov, L.B. *et al.* The repertoire of mutational signatures in human cancer. *Nature* **578**, 94-101 (2020).
30. Baczynska, K., Khazova, M. & O'Hagan, J. Sun exposure of indoor workers in the UK-survey on the time spent outdoors. *Photochemical & Photobiological Sciences* **18**, 120-128 (2019).
31. Lergemuller, S. *et al.* Association of Lifetime Indoor Tanning and Subsequent Risk of Cutaneous Squamous Cell Carcinoma. *JAMA Dermatol* **155**, 1350-1357 (2019).
32. Ziehfrend, S., Schuster, B. & Zink, A. Primary prevention of keratinocyte carcinoma among outdoor workers, the general population and medical professionals: a systematic review updated for 2019. *Journal of the European Academy of Dermatology and Venereology* **33**, 1477-1495 (2019).

18th May 2023

Dear Phil,

How are you?

First, I'm so sorry that it's taken so long to return this decision to you. Thank you so much for your patience.

Thank you for submitting your revised manuscript "Somatic mutations in facial skin from countries of contrasting skin cancer risk" (NG-LE61339R). It has now been seen by the original referees and their comments are below. The reviewers find that the paper has improved in revision, and therefore we will be happy in principle to publish it in Nature Genetics, pending minor revisions to satisfy the referees' final requests (Reviewer #3 has only asked for very minor edits) and to comply with our editorial and formatting guidelines.

** Note that we will send you a checklist detailing these editorial and formatting requirements soon. Please do not finalize your revisions or upload the final materials until you receive this additional information.**

In recognition of the time and expertise our reviewers provide to Nature Genetics's editorial process, we would like to formally acknowledge their contribution to the external peer review of your manuscript entitled "Somatic mutations in facial skin from countries of contrasting skin cancer risk". For those reviewers who give their assent, we will be publishing their names alongside the published article.

While we prepare these instructions, we encourage the Corresponding Author to begin to review and collect the following:

-- Confirmation from all authors that the manuscript correctly states their names, institutional affiliations, funding IDs, consortium membership and roles, author or collaborator status, and author contributions.

-- Declarations of any financial and non-financial competing interests from any author. For the sake of transparency and to help readers form their own judgment of potential bias, the Nature Portfolio require authors to declare any financial and non-financial competing interests in relation to the work described in the submitted manuscript. This declaration must be complete, including author initials, in the final manuscript text.

If you have any questions as you begin to prepare your submission please feel free to contact our Editorial offices at genetics@us.nature.com. We are happy to assist you.

Thank you again for your interest in Nature Genetics.

Sincerely,

Safia Danovi
Editor
Nature Genetics

Reviewer #1 (Remarks to the Author):

The authors have done an excellent job addressing my questions and I have no further comments.

Reviewer #2 (Remarks to the Author):

I have no further questions on the manuscript and thank the authors for their detailed and interesting discussion.

The new data on SCC risk by country is highly interesting and raises many other questions beyond the scope of the current work.

Reviewer #3 (Remarks to the Author):

The authors have thoroughly addressed the concerns of all 3 reviewers, particularly regarding the genetic differences between the two populations and some statistical considerations.

One very minor point that should be easily corrected: for Fig 4, they don't define EA Freq and GBR Freq.

Signed: James DeGregori and Marco De Dominicis

ORCID

Nature Genetics is committed to improving transparency in authorship. As part of our efforts in this direction, we are now requesting that all authors identified as 'corresponding author' create and link their Open Researcher and Contributor Identifier (ORCID) with their account on the Manuscript Tracking System (MTS) prior to acceptance. ORCID helps the scientific community achieve unambiguous attribution of all scholarly contributions. For more information please visit <http://www.springernature.com/orcid>

For all corresponding authors listed on the manuscript, please follow the instructions in the link below to link your ORCID to your account on our MTS before submitting the final version of the manuscript. If you do not yet have an ORCID you will be able to create one in minutes.

IMPORTANT: All authors identified as 'corresponding author' on the manuscript must follow these instructions. Non-corresponding authors do not have to link their ORCIDs but are encouraged to do so. Please note that it will not be possible to add/modify ORCIDs at proof. Thus, if they wish to have their ORCID added to the paper they must also follow the above procedure prior to acceptance.

To support ORCID's aims, we only allow a single ORCID identifier to be attached to one account. If you have any issues attaching an ORCID identifier to your MTS account, please contact the [Platform Support Helpdesk](http://platformsupport.nature.com/).

Final Decision Letter:

6th Jul 2023

Dear Phil,

How are you? I hope you're well.

I am delighted to say that your manuscript "Somatic mutations in facial skin from countries of contrasting skin cancer risk" has been accepted for publication in an upcoming issue of Nature Genetics.

Your paper will be published online after we receive your corrections and will appear in print in the next available issue. You can find out your date of online publication by contacting the Nature Press Office

(press@nature.com) after sending your e-proof corrections. Now is the time to inform your Public Relations or Press Office about your paper, as they might be interested in promoting its publication. This will allow them time to prepare an accurate and satisfactory press release. Include your manuscript tracking number (NG-LE61339R1) and the name of the journal, which they will need when they contact our Press Office.

Please note that *Nature Genetics* is a Transformative Journal (TJ). Authors may publish their research with us through the traditional subscription access route or make their paper immediately open access through payment of an article-processing charge (APC). Authors will not be required to make a final decision about access to their article until it has been accepted. [Find out more about Transformative Journals](https://www.springernature.com/gp/open-research/transformative-journals)

Authors may need to take specific actions to achieve [compliance](https://www.springernature.com/gp/open-research/funding/policy-compliance-faqs) with funder and institutional open access mandates. If your research is supported by a funder that requires immediate open access (e.g. according to [Plan S principles](https://www.springernature.com/gp/open-research/plan-s-compliance)) then you should select the gold OA route, and we will direct you to the compliant route where possible. For authors selecting the subscription publication route, the journal's standard licensing terms will need to be accepted, including [self-archiving and license to publish](https://www.nature.com/nature-portfolio/editorial-policies/self-archiving-and-license-to-publish). Those licensing terms will supersede any other terms that the author or any third party may assert apply to any version of the manuscript.

Please note that Nature Portfolio offers an immediate open access option only for papers that were first submitted after 1 January, 2021.

If you have not already done so, we invite you to upload the step-by-step protocols used in this manuscript to the Protocols Exchange, part of our on-line web resource, natureprotocols.com. If you complete the upload by the time you receive your manuscript proofs, we can insert links in your article that lead directly to the protocol details. Your protocol will be made freely available upon publication of your paper. By participating in natureprotocols.com, you are enabling researchers to more readily reproduce or adapt the methodology you use. [Natureprotocols.com](http://natureprotocols.com) is fully searchable, providing your protocols and paper with increased utility and visibility. Please submit your protocol to <https://protocolexchange.researchsquare.com/>. After entering your nature.com username and password you will need to enter your manuscript number (NG-LE61339R1). Further information can be found at <https://www.nature.com/nature-portfolio/editorial-policies/reporting-standards#protocols>

Sincerely,

Safia Danovi
Editor
Nature Genetics